# Bridging the Gap: Providing Post-Hoc Symbolic Explanations for Sequential Decision-Making Problems with Inscrutable Representations

**Sarath Sreedharan, Utkarsh Soni, Mudit Verma, Siddharth Srivastava, Subbarao Kambhampati**
School of Computing & AI, Arizona State University
{ssreedh3, usoni1, mverma13, siddharths, rao}@asu.edu

## Abstract

As increasingly complex AI systems are introduced into our daily lives, it becomes important for such systems to be capable of explaining the rationale for their decisions and allowing users to contest these decisions. A significant hurdle to allowing for such explanatory dialogue could be the *vocabulary mismatch* between the user and the AI system. This paper introduces methods for providing contrastive explanations in terms of user-specified concepts for sequential decision-making settings where the system's model of the task may be best represented as an inscrutable model. We do this by building partial symbolic models of a local approximation of the task that can be leveraged to answer the user queries. We test these methods on a popular Atari game (Montezuma's Revenge) and variants of Sokoban (a well-known planning benchmark) and report the results of user studies to evaluate whether people find explanations generated in this form useful.

## 1 Introduction

As AI systems are increasingly being deployed in scenarios involving high-stakes decisions, the issue of *interpretability* of their decisions to the humans in the loop has acquired renewed urgency. Most work on interpretability has hither-to focused on one-shot classification tasks, and has revolved around ideas such as marking regions of the input instance (e.g. image) that were most salient in the classification of that instance (Selvaraju et al., 2016; Sundararajan et al., 2017). Such methods used for one-shot classification tasks don't generalize well to sequential decision tasks–that are the focus of this paper. In particular, in sequential decision tasks, humans in the loop might ask more complex "contrastive" explanatory queries–such as why the system took one particular course of action instead of another. The AI system's answers to such questions will often be tied to its internal representations and reasoning, traces of which are not likely to be comprehensible as explanations to humans in the loop. Some have viewed this as an argument to make AI systems use reasoning over symbolic models. We don't take a position on the methods AI systems use to arrive at their decisions. We do however think that the onus of making their decisions interpretable to humans in the loop *in their own vocabulary* should fall squarely on the AI systems. Indeed, we believe that orthogonal to the issue of whether AI systems use internal

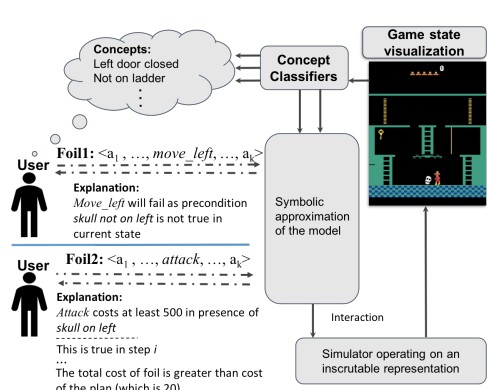

Figure 1: Explanatory dialogue starts when the user presents a specific alternate plan. The system explains the preference over this alternate plan in terms of model information expressed in terms of propositional concepts specified by the user, operationalized as classifiers.

symbolic representations to guide their own decisions, they need to develop local symbolic models that are interpretable to humans in the loop, and use them to provide explanations for their decisions.

Accordingly, in this work, we develop a framework that takes a set of previously agreed-upon user vocabulary terms and generates contrastive explanations in these terms Miller (2018); Hesslow (1988). Specifically, we will do this by learning components of a local symbolic dynamical model (such as PDDL (McDermott et al., 1998)) that captures the agent model in terms of actions with preconditions and effects from the user-specified vocabulary. There is evidence that such models conform to folk psychology Malle (2006). This learned local model is then used to explain the potential infeasibility or suboptimality of the alternative raised by the user. Figure 1 presents the flow of the proposed explanation generation process in the context of Montezuma's Revenge (Wikipedia contributors, 2019). In this paper, we will focus on deterministic settings (Section 3), though we can easily extend the ideas to stochastic settings, and will ground user-specified vocabulary terms in the AI system's internal representations via learned classifiers (Section 4). The model components required for explanations are learned by using experiences (i.e. state-action-state sets) sampled from the agent model (Section 4 & 5). Additionally, we formalize the notion of local symbolic approximation for sequential decision-making models, and introduce the idea of explanatory confidence (Section 6). The exaplanatory confidence captures the fidelity of explanations and help ensure the system only provides explanation whose confidence is above a given threshold. We will evaluate the effectiveness of the method through both systematic (IRB approved) user studies and computational experiments (Section 7). As we will discuss in Section 2, our approach has some connections to (Kim et al., 2018), which while focused on one-shot classification tasks, also advocates explanations in terms of concepts that have meaning to humans in the loop. Our approach of constructing local model approximations is akin to LIME (Ribeiro et al., 2016), with the difference being that we construct symbolic dynamical models in terms of human vocabulary, while LIME constructs approximate linear models over abstract features that are *machine generated*.

## 2 RELATED WORK

The representative works in the direction of using concept level explanation include works like (Bau et al., 2017), TCAV (Kim et al., 2018) and its various offshoots like (Luss et al., 2019) that have focused on one-shot decisions. These works take a line quite similar to us in that they try to create explanations for current decisions in terms of a set of user-specified concepts. While these works don't explicitly reason about explanatory confidence they do discuss the possibility of identifying inaccurate explanation, and tries to address them through statistical tests. Another thread of work, exemplified by works like (Koh et al., 2020; Lin et al., 2020), tries to force decisions to be made in terms of user-specified concepts, which can then be used for explanations. There have also been recent works on automatically identifying human-understandable concepts like Ghorbani et al. (2019) and Hamidi-Haines et al. (2018). We can also leverage these methods when our system identifies scenarios with insufficient vocabulary.

Most works in explaining sequential decision-making problems either use a model specified in a shared vocabulary as a starting point for explanation or focus on saliency-based explanations (cf. (Chakraborti et al., 2020)), with very few exceptions. Authors of (Hayes & Shah, 2017) have looked at the use of high-level concepts for policy summaries. They use logical formulas to concisely characterize various policy choices, including states where a specific action may be selected (or not). Unlike our work, they are not trying to answer why the agent chooses a specific action (or not). (Waa et al., 2018) looks at addressing the suboptimality of foils while supporting interpretable features, but it requires the domain developer to assign positive and negative outcomes to each action. In addition to not addressing possible vocabulary differences between a system developer and the end-user, it is also unclear if it is always possible to attach negative and positive outcomes to individual actions.

Another related work is the approach studied in (Madumal et al., 2020). Here, they are also trying to characterize dynamics in terms of high-level concepts but assume that the full structural relationship between the various variables is provided upfront. The explanations discussed in this paper can also be seen as a special case of Model Reconciliation explanation (c.f (Chakraborti et al., 2017)), where the human model is considered to be empty. The usefulness of preconditions as explanations has also been studied by works like (Winikoff, 2017; Broekens et al., 2010). Our effort to associate action cost to concepts could also be contrasted to efforts in (Juozapaitis et al., 2019) and (Anderson et al., 2019) which leverage interpretable reward components. Another group of works popular in

RL explanations is the ones built around saliency maps (Greydanus et al., 2018; Iyer et al., 2018; Puri et al., 2019), which tend to highlight parts of the state that are important for the current decision. In particular, we used (Greydanus et al., 2018) as a baseline because many follow-up works have shown their effectiveness (cf. (Zhang et al., 2020)). Readers can refer to Alharin et al. (2020) for a recent survey of explanations in RL.

Another related thread of work, is that of learning models (some representative works include (Carbonell & Gil, 1990; Stern & Juba, 2017; Wu et al., 2007)). To the best of our knowledge none of the works in this direction allow for noisy observations of state and none focused on identifying specific model components. While we are unaware of any works that provide confidence over learned model components, (Stern & Juba, 2017) provides loose PAC guarantees over the entire learned model.

## 3 PROBLEM SETTING

Our setting consists of an agent, be it programmed or RL-based, that has a model of the dynamics of the task that is inscrutable to the user (in so far that the user can't directly use the model representation used by the agent) in the loop and uses a decision-making process that is sound for this given model. Note that here the term *model* is being used in a very general sense. These could refer to tabular models defined over large atomic state spaces, neural network-based dynamics models possibly learned over latent representation of the states and even simulators. The only restriction we place on the model is that we can sample possible experiences from it. Regardless of the true representation, we will denote the model by the tuple $\mathcal{M} = \langle S, A, T, \mathcal{C} \rangle$. Where $S$ and $A$ are the state and action sets and $T : S \times A \rightarrow S \cup \{\bot\}$ ($\bot$ the absorber failure state) and $\mathcal{C} : S \times A \rightarrow \mathbb{R}$ capture the transition and cost function. We will use $\bot$ to capture failures that could occur when the agent violates hard constraints like safety constraints or perform any invalid actions. We will consider goal-directed agents that are trying to drive the state of the world to one of the goal states ($\mathbb{G}$ being the set) from an initial state $I$. The solution takes the form of a sequence of actions or a plan $\pi$.

We will use symbolic action models with preconditions and cost functions (similar to PDDL models (Geffner & Bonet, 2013)) as a way to approximate the problem for explanations. Such a model can be represented by the tuple $\mathcal{M}_\mathcal{S} = \langle F_\mathcal{S}, A_\mathcal{S}, I_\mathcal{S}, G_\mathcal{S}, \mathcal{C}_\mathcal{S} \rangle$, where $F_\mathcal{S}$ is a set of propositional state variables defining the state space, $A_\mathcal{S}$ is the set of actions, $I_\mathcal{S}$ is the initial state, $G_\mathcal{S}$ is the goal specification. Each valid problem state is uniquely identified by the subset of state variables that are true in that state (so for any state $s \in S_{\mathcal{M}_\mathcal{S}}$, where $S_{\mathcal{M}_\mathcal{S}}$ is the set of states for $\mathcal{M}_\mathcal{S}$, $s \subseteq F_\mathcal{S}$). Each action $a \in A_\mathcal{S}$ is further described in terms of the preconditions $prec_a$ (specification of states in which $a$ is executable) and the effects of executing the action. We will denote the state formed by executing action $a$ in a state $s$ as $a(s)$. We will focus on models where the preconditions are represented as a conjunction of propositions. If the action is executed in a state with missing preconditions, then the execution results in the invalid state $\bot$. Unlike standard STRIPS models, where the cost of executing action is independent of states, we will use a state-dependent cost function of the form $\mathcal{C}_\mathcal{S} : 2^F \times A_\mathcal{S} \rightarrow \mathbb{R}$ to capture forms of cost functions popular in RL benchmarks.

## 4 CONTRASTIVE EXPLANATIONS

The specific explanatory setting, illustrated in Figure 1, is one where the agent comes up with a plan $\pi$ (to achieve one of the goals specified in $\mathbb{G}$ from $I$) and the user responds by raising an alternate plan $\pi_f$ (*the foil*) that they believe should be followed instead. Now the system needs to explain why $\pi$ may be preferred over $\pi_f$, by showing that $\pi_f$ is invalid (i.e., $\pi_f$ doesn't lead to a goal state or one of the action in $\pi_f$ results in the invalid state $\bot$) or $\pi_f$ is costlier than $\pi$ ($\mathcal{C}(I, \pi) < \mathcal{C}(I, \pi_f)$).[1]

To concretize this interaction, consider the modified version of Montezuma's Revenge (Figure 1). The agent starts from the highest platform, and the goal is to get to the key. The specified plan $\pi$ may require the agent to make its way to the lowest level, jump over the skull, and then go to the key with a total cost of 20. Now the user raises two possible foils that are similar to $\pi$ but use different strategies in place of jumping over the skull. *Foil1*: instead of jumping, the agent just moves left (i.e., it tries to move *through* the skull) and, *Foil2*: instead of jumping over the skull, the agent performs the attack action (not part of the original game, but added here for illustrative purposes) and then moves on to the key. Using the internal model, the system can recognize that in the first case, moving left would lead to an invalid state, and in the second case, the foil is costlier, though effectively communicating this to the user is a different question. If there exists a shared

---

[1]If the foil is as good as the original plan or better, then the system could switch to the foil.

visual communication channel, the agent could try to demonstrate the outcome of following these alternate strategies. Unfortunately, this would not only necessitate additional cognitive load on the user's end to view the demonstration, but it may also be confusing to the user in so far that they may not be able to recognize why in a particular state the move left action was invalid and attack action costly. As established in our user study and pointed out by Atrey et al. (2019), even highlighting of visual features may not effectively resolve this confusion. This scenario thus necessitates the use of methods that are able to express possible explanations in terms that the user may understand.

**Learning Concept Maps:** The input to our system is *the set of propositional concepts the user associates with the task*. For Montezuma, this could involve concepts like *agent being on a ladder, holding onto the key, being next to the skull*. Each concept corresponds to a propositional fact that the user associates with the task's states and believes their presence or absence in a state could influence the dynamics and the cost function. We can collect such concepts from subject matter experts as by Cai et al. (2019), or one could just let the user interact with or observe the agent and then provide a possible set of concepts. We used the latter approach to collect the propositions for our evaluation of the Sokoban domains (Section 7 and A.7). Each concept corresponds to a binary classifier, which detects whether the proposition is present or absent in a given internal state (thus allowing us to convert the atomic states into a factored representation). Let $\mathbb{C}$ be the set of classifiers corresponding to the high-level concepts. For state $s \in S$, we will overload the notation $\mathbb{C}$ and specify the concepts that are true as $\mathbb{C}(s)$, i.e., $\mathbb{C}(s) = \{c_i | c_i \in \mathbb{C} \wedge c_i(s) = 1\}$ (where $c_i$ is the classifier corresponding to the $i^{th}$ concept and we overload the notation to also stand for the label of the $i^{th}$ concept). The training set for such concept classifiers could come from the user (where they provide a set of positive and negative examples per concept). Classifiers can be then learned over the model states or the internal representations used by the agent decision-making (for example activations of intermediate neural network layers).

**Explanation using concepts:** To explain the preference of plan $\pi$ over foil $\pi_f$, we will present model information to the user taken from a symbolic representation of the agent model. But rather than requiring this model to be an exact representation of the complete agent model, we will instead focus on accurately capturing a subset of the model by instead trying to learn a local approximation

**Definition 1** *A symbolic model* $\mathcal{M}_S^{\mathbb{C}} = \langle \mathbb{C}, A_S^{\mathbb{C}}, \mathbb{C}(I), \mathbb{C}(\mathbb{G}), \mathcal{C}_S^{\mathbb{C}} \rangle$. *is said to be a* local symbolic approximation *of the model* $\mathcal{M}^R = \langle S, A, T, \mathcal{C} \rangle$ *for regions of interest* $\hat{S} \subseteq S$ *if* $\forall s \in \hat{S}$ *and* $\forall a \in A$, *we have an equivalent action* $a^{\mathbb{C}} \in A_S^{\mathbb{C}}$, *such that (a)* $a^{\mathbb{C}}(\mathbb{C}(s)) = \mathbb{C}(T(s,a))$ *(assuming* $\mathbb{C}(\perp) = \perp$*) and (b)* $\mathcal{C}_S^{\mathbb{C}}(\mathbb{C}(s), a) = \mathcal{C}(s, a)$ *and (c)* $\mathbb{C}(\mathbb{G}) = \bigcap_{s_g \in \mathbb{G} \cap \hat{S}} \mathbb{C}(s_g)$.

Following Section 3, this is a PDDL-style model with preconditions and conditional costs defined over the conjunction of positive propositional literals. A.2 establishes the sufficiency of this representation to capture arbitrarily complex preconditions (including disjunctions) and cost functions expressed in terms of the proposition set $\mathbb{C}$. Also to establish the preference of plan does not require informing the users about the entire model $\mathcal{M}_S^{\mathbb{C}}$, but rather only the relevant parts. To establish the invalidity of $\pi_f$, we only need to explain the failure of the first failing action $a_i$, i.e., the one that resulted in the invalid state (for *Foil1* this corresponds to move-left action at the state visualized in Figure 1). We explain the failure of action in a state by pointing out a proposition that is an action precondition which is absent in the given state. Thus a concept $c_i \in \mathbb{C}$ is considered an explanation for failure of action $a_i$ at state $s_i$, if $c_i \in prec_{a_i} \setminus \mathbb{C}(s_i)$. For *Foil1*, the explanation would be – ***the action move-left failed in the state as the precondition skull-not-on-left was false in the state***.

This formulation can also capture failure to achieve goal by appending an additional goal action at the end of the plan, which causes the state to transition to an end state, and fails for all states except the ones in $\mathbb{G}$. Note that instead of identifying all the missing preconditions, we focus on identifying a single precondition, as this closely follows prescriptions from works in social sciences that have shown that selectivity or minimality is an essential property of effective explanations (Miller, 2018).

For explaining suboptimality, we inform the user about the symbolic cost function $\mathcal{C}_S^{\mathbb{C}}$. To ensure minimality, rather than provide the entire cost function, we will instead try to learn and provide an abstraction of the cost function $\mathcal{C}_s^{abs}$

**Definition 2** *For the symbolic model* $\mathcal{M}_S^{\mathbb{C}} = \langle \mathbb{C}, A_S^{\mathbb{C}}, \mathbb{C}(I), \mathbb{C}(\mathbb{G}), \mathcal{C}_S^{\mathbb{C}} \rangle$, *an abstract cost function* $\mathcal{C}_S^{abs} : 2^{\mathbb{C}} \times A_S^{\mathbb{C}} \rightarrow \mathbb{R}$ *is specified as:* $\mathcal{C}_S^{abs}(\{c_1, .., c_k\}, a) = min\{\mathcal{C}_S^{\mathbb{C}}(s, a) | s \in S_{\mathcal{M}_S^{\mathbb{C}}} \wedge \{c_1, .., c_k\} \subseteq s\}$.

| **a** Algorithm for Finding Missing Precondition | **b** Algorithm for Finding Cost Function |
|---|---|
| 1: **procedure** PRECONDITION-SEARCH | 1: **procedure** COST-FUNCTION-SEARCH |
| 2:   *Input*:  $s_{\text{fail}}, a_{\text{fail}}$, Sampler, $\mathcal{M}^R, \mathbb{C}, \ell$ | 2:   *Input*:  $\pi_f, \mathcal{C}_\pi$, Sampler, $\mathcal{M}^R, \mathbb{C}, \ell$ |
| 3:   *Output*: Missing precondition $C_{\text{prec}}$ | 3:   *Output*: $\mathbb{C}_{\pi_f}$ |
| 4:   *Procedure*: | 4:   *Procedure*: |
| 5:     $\mathcal{H}_{\mathbb{P}} \leftarrow$ Missing concepts in $s_{fail}$ | 5:     **for** conc_limit in 1 to $|\mathbb{C}|$ **do** |
| 6:     sample_count = 0 | 6:       current_foil_cost = 0, conc_list = [], $s \leftarrow I$ |
| 7:     **while** sample_count $< \ell$ **do** | 7:       **for** i in 1 to k (the length of the foil) **do** |
| 8:       $s \sim$ Sampler | 8:         $s \leftarrow T(s, a_i)$ |
| 9:       **if** $T(s, a_{\text{fail}}) \neq \bot$ **then** | 9:         $\hat{\mathbb{C}}_i$, min_cost = |
| 10:         $\mathcal{H}_{\mathbb{P}} \leftarrow \mathbb{C}(s) \cap \mathcal{H}_{\mathbb{P}}$ | 10: find_min_conc_set($\mathbb{C}(s), a_i$, conc_limit, $\ell$) |
| 11:         **if** $|\mathcal{H}_{\mathbb{P}}| = 0$ **then return** Signal that concept list is incomplete | 11:         current_foil_cost += min_cost |
| | 12:         conc_list.push($\hat{\mathbb{C}}_i$, min_cost) |
| 12:       sample_count += 1 | 13:       **if** current_foil_cost $> \mathcal{C}_\pi$ **then return** conc_list |
| **return** any $C_i \in$ poss_prec_set | **return** Signal that the concept list is incomplete |

Figure 2: Algorithms for identifying (a) missing precondition and (b) cost function

Intuitively, $\mathcal{C}_\mathcal{S}^{abs}(\{c_1, .., c_k\}, a) = k$ can be understood as stating that *executing the action a, in the presence of concepts* $\{c_1, .., c_k\}$ *costs at least k*. We can use $\mathcal{C}_\mathcal{S}^{abs}$ in an explanation by identifying a sequence of concept set $\mathbb{C}_{\pi_f} = \langle \hat{\mathbb{C}}_1, ..., \hat{\mathbb{C}}_k \rangle$, corresponding to each step of the foil $\pi_f = \langle a_1, .., a_k \rangle$, such that (a) $\hat{\mathbb{C}}_k$ is a subset of concepts in the corresponding state reached by the foil and (b) the total cost of abstract cost function defined over the concept subsets are larger than the plan cost $\sum_{i=\{1..k\}} \mathcal{C}_\mathcal{S}^{abs}(\hat{\mathbb{C}}_i, a_i) > \mathcal{C}(I, \pi)$. For *Foil2*, the explanation would include the information – *executing the action **attack** in the presence of the concept **skull-on-left**, will cost at least 500*.

## 5  IDENTIFYING EXPLANATIONS THROUGH SAMPLE-BASED TRIALS

For identifying the model parts, we will rely on the internal model to build symbolic estimates. Since we can separate the two explanation cases using the agent's internal model, we will only focus on the problem of identifying the model parts given the required explanation type.

**Identifying failing precondition:** The first case we will consider is the one related to explaining plan failures. In particular, given the failing state $s_{\text{fail}}$ and failing foil action $a_{\text{fail}}$, we want to find a concept that is absent in $s_{\text{fail}}$ but is a precondition for the action $a_{\text{fail}}$. We will try to approximate whether a concept is a precondition or not by checking whether they appear in all the states sampled from the model, where $a_{\text{fail}}$ is executable ($\mathbb{S}$ - set of all sampled states). Figure 2(a) presents the pseudo-code for finding such preconditions. We rely on the sampler (denoted as Sampler) to create the set $\mathbb{S}$, where the number of samples used is upper-bounded by a sampling budget $\ell$. poss_prec_set captures the set of hypotheses regarding the missing precondition maintained over the course of the search.

We ensure that models learned are local approximations by considering only samples within some distance from the states in the plan and foils. For reachability-based distances, we can use random walks to generate the samples. Specifically, we can start a random walk from one of the states observed in the plan/foil and end the sampling episode whenever the number of steps taken crosses a given threshold. The search starts with a set of hypotheses for preconditions $\mathcal{H}_{\mathbb{P}}$ and rejects individual hypotheses whenever the search sees a state where the action $a_{fail}$ is executable and the concept is absent. The worst-case time complexity of this search is linear on the sampling budget.

If the hypotheses set turns empty, *it points to the fact that the current vocabulary set is insufficient to explain the given plan failure*. Focusing on a single model-component not only simplifies the learning problem but as we will see in Section 6 allows us to quantify uncertainty related to the learned model components and also allow for noisy observation. These capabilities are missing from previous works.

**Identifying cost function:** We will employ a similar sampling-based method to identify the cost function abstraction. Unlike the precondition failure case, there is no single action we can choose, but rather we need to choose a level of abstraction for each action in the foil (though it may be possible in many cases to explain the suboptimality of foil by only referring to a subset of actions in the foil). For the concept subset sequence ($\mathbb{C}_{\pi_f} = \langle \hat{\mathbb{C}}_1, ..., \hat{\mathbb{C}}_k \rangle$) that constitutes the explanation, we

will also try to minimize the total number of concepts used in the explanation ($\sum_{i=1..k} \|\hat{\mathbb{C}}_i\|$). Figure 2(b) presents a greedy algorithm to find such cost abstractions. The procedure find_min_conc_set, takes the current concept representation of state $s_i$ in the foil and searches for the subset $\hat{\mathbb{C}}_i$ of $\mathbb{C}(s_i)$ with the maximum value for $\mathcal{C}_{\mathcal{S}}^{abs}(\hat{\mathbb{C}}_i, a_i)$, where the value is approximated through sampling (with budget $\ell$), and the subset size is upper bounded by conc_limit. As mentioned in Definition 2, the value of $\mathcal{C}_{\mathcal{S}}^{abs}(\hat{\mathbb{C}}_i, a_i)$ is given as the minimal cost observed when executing the action $a_i$ in a state where $\hat{\mathbb{C}}_i$ are true. The algorithm incrementally increases the conc_limit value till a solution is found or if it crosses the vocabulary set size. Note that this algorithm is not an optimal one, but we found it to be effective enough for the scenarios we tested. We can again enforce the required locality within the sampler and similar to the previous case, *we can identify the insufficiency of the concept set by the fact that we aren't able to identify a valid explanation when conc_limit is at vocabulary size*. The worst case time complexity of the search is $\mathcal{O}(|\mathbb{C}| \times |\pi_f| \times |\ell|)$. We are unaware of any existing works for learning such abstract cost-functions.

## 6 QUANTIFYING EXPLANATORY CONFIDENCE

One of the critical challenges faced by any post hoc explanations is the question of fidelity and how accurately the generated explanations reflect the decision-making process. In our case, we are particularly concerned about whether the model component learned using the algorithms mentioned in Section 5 is part of the exact symbolic representation $\mathcal{M}_{\mathcal{S}}^{\mathbb{C}}$ (for a given set of states $\hat{S}$ and actions $\hat{A}$). Given that we are identifying model components based on concepts provided by the user, it would be easy to generate explanations that feed into the user's confirmation biases about the task.

Our confidence value associates a probability of correctness with each learned model component by leveraging learned relationships between the concepts. Additionally, our confidence measure also captures the fact that the learned concept classifiers will be noisy at best. So we will use the output of the classifiers as noisy observations of the underlying concepts, with the observation model $P(O_{c_i}^s | c_i \in S)$ (where $O_{c_i}^s$ captures the fact that the concept was observed and $c_i \in S$ represents whether the concept was present in the state) defined by a probabilistic model of the classifier prediction.

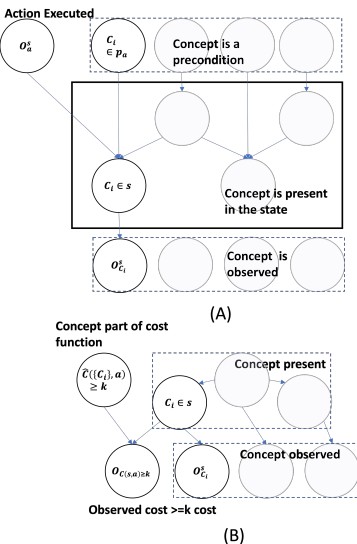

Figure 3: Simplified probabilistic graphical models for confidence of learned (A) Preconditions and (B) Cost.

Figure 3, presents a graphical model (with notations defined inline) for measuring the posterior of a model component being true given an observation generated from the classifier on a sampled state. For a given observation of executions of action $a$ in state $s$ ($O_a^s = True$ or just $O_a^s$), the positive or negative observation of a concept $c_i$ ($O_{c_i}^s = x$, where $x \in \{True, False\}$) and an observation that action's cost is greater than $k$ ($O_{\mathcal{C}(s,a) \geq k}$), the updated explanatory confidence for each explanation type is provided as (1) $P(c_i \in p_a | O_{c_i}^s = x \wedge O_a^s \wedge O_{\mathbb{C}(s) \setminus c_i})$, where $c_i \in p_a$ captures the fact that the concept is part of the precondition of $a$ and $O_{\mathbb{C}(s) \setminus c_i}$ is the observed status of the rest of the concepts, and (2) $P(\mathcal{C}_s^{abs}(\{c_i\}, a) \geq k | O_{c_i}^s = x \wedge O_{\mathcal{C}(s,a)>=k} \wedge O_{\mathbb{C}(s) \setminus c_i})$, where $\mathcal{C}_s^{abs}(\{c_i\}, a) \geq k$ asserts that the abstract cost function defined over concept $c_i$ is greater than $k$

The final probability for the explanation would be the posterior calculated over all the state samples. Additionally, we can also extend these probability calculations to multi-concept cost functions. We can now incorporate these confidence measures directly into the search algorithms described in Section 5 to find the explanation that maximizes these probabilities. In the case of precondition identification, given a noisy classifier, we can no longer use the classifier output to directly remove a concept from the hypothesis set $\mathcal{H}_{\mathbb{P}}$. At the start of the search, we associate a prior to each hypothesis and use the observed concept value at each executable state to update the posterior. We only remove a precondition from consideration if the probability associated with it dips below a threshold $\kappa$.

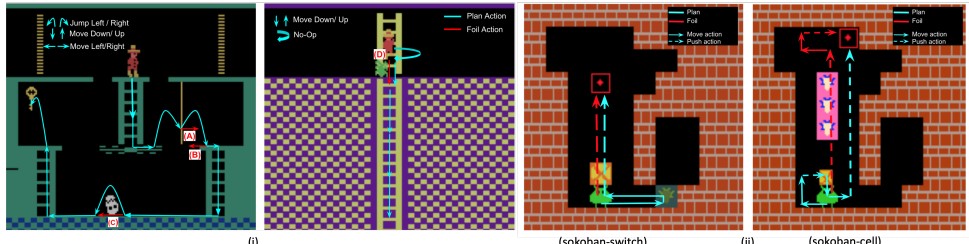

Figure 4: The plan and foils used in the evaluation, plans are highlighted in blue and foils in red: (i) shows the two montezuma's revenge screens used (ii) the two variations of the sokoban game used. For Montezuma the concepts corresponding to missing precondition are not_on_rope, not_on_left_ledge, not_skull_on_left and is_clear_down_of_crab for failure points A, B, C and D respectively (note all concepts were originally defined as positives and we formed the not_ concepts by negating the output of the classifier). Concepts switch_on and on_pink_cell for Sokoban cost variants that results in push action costing 10 (instead of 1)

For cost function identification, if we have multiple cost functions with the same upper bound, we select the one with the highest probability. We also allow for noisy observations in the case of cost identification. The updated pseudo codes are provided in A.3.

We can simplify the probability calculations by making the following assumptions: (1) the distribution of all non-precondition concepts in states where the action is executable is the same as their overall distribution (which can be empirically estimated), (2) The cost distribution of action over states corresponding to a concept that does not affect the cost function is identical to the overall distribution of cost for the action (which can again be empirically estimated). The first assumption implies that the likelihood of seeing a non-precondition concept in a sampled state where the action is executable is equal to the likelihood of it appearing in any sampled state. In the most general case, this equals $P(c_i \in s | O_{\mathbb{C}(s) \setminus c_i})$, i.e., the likelihood of seeing this concept given the other concepts in the state. The second assumption implies that for a concept that has no bearing on the cost function for an action, the likelihood that executing the action in a state where the concept is present will result in a cost greater than $k$ will be the same as that of the action execution resulting in a cost greater than $k$ for a randomly sampled state ($p_{\mathcal{C}(\cdot,a) \geq k}$). This assumption can be further relaxed by considering the distribution of the action cost given the rest of the observed set of concepts (though this would require more samples to learn). *The full derivation of the probability calculations with the assumptions is provided in A.4 and an empirical evaluation of the assumptions in A.10. All results reported in the next section were calculated using the probabilistic version of the search while allowing for noisy concept maps.*

## 7 EVALUATION

We tested the approach on the open-AI gym's deterministic implementation of Montezuma's Revenge (Brockman et al., 2016) for precondition identification and two modified versions of the gym implementation of Sokoban (Schrader, 2018) for both precondition and cost function identification. *To simplify calculations for the experiments, we made an additional assumption that the concepts are independent.* All hyperparameters used by learning algorithms are provided in A.6.

**Montezuma's Revenge:** We used RAM-based state representation here. To introduce richer preconditions to the settings, we marked any non-noop action that fails to change the current agent state and action that leads to midair states where the agent is guaranteed to fall to death (regardless of actions taken) as failures. We selected four invalid foils (generated by the authors by playing the game), three from screen-1 and one from screen-4 of the game (Shown in Figure 4 (i)). The plan for screen-1 involved the player reaching a key and for screen-2 the player has to reach the bottom of the screen. We specified ten concepts, which we believed would be relevant to the game, for each screen and collected positive and negative examples using automated scripts. We used AdaBoost Classifiers (Freund et al., 1999) for the concepts and had an average accuracy of 99.72%.

**Sokoban Variants:** The original sokoban game involves the player pushing a box from one position to a target position. We considered two variations of this basic game. One that requires a switch (green cell) the player could turn on before pushing the box (referred to as Sokoban-switch), and a second version (Sokoban-cells) included particular cells (highlighted in pink) from which it is

costlier to push the box. For Sokoban-switch, we had two variations, one in which turning on the switch was a precondition for push actions and another one in which it merely reduced the cost of pushing the box. The plan and foil (Shown in Figure 4 (ii)) were generated by the authors by playing the game. *We used a survey of graduate students unfamiliar with our research to collect the set of concepts for these variants*. The survey allowed participants to interact with the game through a web interface (the cost-based version for Sokoban-switch and Sokoban-cell), and at the end, they were asked to describe game concepts that they thought were relevant for particular actions. We received 25 unique concepts from six participants for Sokoban-switch and 38 unique concepts from seven participants for Sokoban-cell. We converted the user descriptions of concepts to scripts for sampling positive and negative instances. We focused on 18 concepts and 32 concepts for Sokoban-switch and Sokoban-cell, respectively, based on the frequency with which they appear in game states, and used Convolutional Neural Networks (CNNs) for the classifier (which mapped state images to concepts). The classifiers had an average accuracy of 99.46% (Sokoban-switch) and 99.34% (Sokoban-cell).

**Computational Experiments:** We ran the search to identify preconditions for Montezuma's foils and Sokoban-switch and cost function for both Sokoban variants. From the original list of concepts, we doubled the final concept list used by including negations of each concept (20 each for Montezuma and 36 and 64 for Sokoban variants). The probabilistic models for each classifier were calculated from the corresponding test sets. For precondition identification, the search was run with a cutoff probability of 0.01 for each concept. In each case, our search-based methods were able to identify the correct model component for explanations. Figure 5(A), presents the probability our system assigns to the correct missing precondition plotted against the sampling budget. It presents the average calculated across ten random episodes (which were seeded by default using urandom (Linux, 2013)). We see that in general, the probability increases with the sampling budget with a few small dips due to misclassifications. We had the smallest explanation likelihood for foil1 in screen 1 in Montezuma (with $0.511 \pm 0.001$), since it used a common concept, and its presence in the executable states was not strong evidence for them being a precondition. Though even in this case, the system correctly identified this concept as the best possible hypothesis for precondition as all the other hypotheses were eliminated after around 100 samples. Similarly, Figure 5(B) plots the probability assigned to the abstract cost functions.

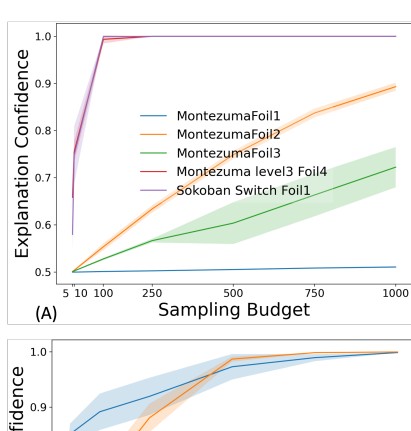

(A)

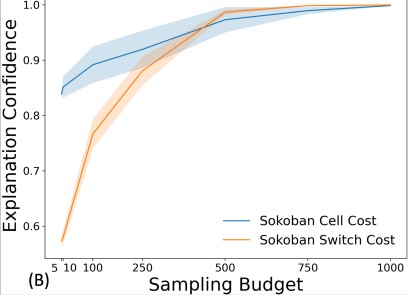

(B)

Figure 5: The average probability assigned to the correct model component by the search algorithms, calculated over ten random search episodes with std-deviation.

**User study:** With the basic explanation generation methods in place, we were interested in evaluating if users would find such an explanation helpful. All study designs followed our local IRB protocols. We were interested in measuring the effectiveness of the symbolic explanations over two different dimensions: (a) whether people prefer explanations that refer to the symbolic model components over a potentially more concise and direct explanation and (b) whether such explanations help people get a better understanding of the task (this mirrors the recommendation established by works like Hoffman et al. (2018)). All study participants were graduate students (different from those who specified the concepts), the demographics of participants can be found in A.9.

For measuring the preference, we tested the hypothesis
*H1: People would prefer explanations that establish the corresponding model component over ones that directly presents the foil information (i.e. the failing action and per-step cost)*

This is an interesting hypothesis, as in our case, the explanatory text for the baseline is a lot more concise (in terms of text-size). As per the selectivity principle (Miller, 2018), people prefer explanations that doesn't contain any extraneous information and thus we are effectively testing here whether people find the model information extraneous. The exact screenshots of the conditions

Table 1: Results from the user study

| | Prefers symbols | Average Likert-score | P-value | | Method | # of Participant | Average Time Taken (sec) | Average # of Steps |
|---|---|---|---|---|---|---|---|---|
| Precondition | 19/20 | 3.47 | $1.0 \times 10^{-8}$ | | Concept-Based | 23 | $43.78 \pm 12.59$ | $35.87 \pm 9.69$ |
| Cost | 16/20 | 3.21 | 0.03 | | Saliency Map | 25 | $134.24 \pm 61.72$ | $52.64 \pm 11.11$ |

(a) H1                                                                        (b) H2

are provided in A.12. We used a *within-subject* study design where the participants were shown an explanation generated by our method along with a simple baseline. Precondition case involved pointing out the failing action and the state it was executed and for the cost case the exact cost of executing each action in the foil was presented. The users were asked to choose the one they believed was more useful (*the choice ordering was randomized to ensure the results were counterbalanced*) and were also asked to report on a five-point Likert scale the completeness of the chosen explanation (1 being not at all complete and 5 being complete). For precondition, we collected 5 responses per each Montezuma foil, and for cost we did 10 per each sokoban variant (40 in total). Table 1a, presents the summary of results from the user study and supports H1. In fact, a binomial test shows that the selections are statistically significant with respect to $\alpha = 0.05$.

For testing the effectiveness of explanation in helping people understand the task, we studied
***H2****: Concept-based precondition explanations help users understand the task better than saliency map based ones.*

We focused saliency maps and preconditions, as saliency map based explanation could highlight areas corresponding to failed precondition concepts (especially when concepts correspond to local regions within the image). *Currently we are unaware of any other existing explanation generation method which can generate explanation for this scenario without introducing additional knowledge about the task*. We measured the user's understanding of the task by their ability to solve the task by themselves. We used a between-subject study design, where each participant was limited to a single explanation type. Each participant was allowed to play the precondition variant of the sokoban-switch game. They were asked to finish the game within 3 minutes and were told that there would be bonuses for people who finish the game in the shortest time. They were not provided any exact instructions about the dynamics of the game. During the game, if the participant performs an action whose preconditions are not met, they are shown their respective explanation. Additionally, the current episode ends and they will have to restart the game from the beginning. One group of users were provided the precondition explanations generated through our method, while the rest were presented with a saliency map. For the latter, we used a state of the art saliency-map based explanation method Greydanus et al. (2018). The participants were told that highlighted regions are parts of the state an AI agent would focus on if it was acting in that state. In all the saliency map explanations, the highlighted region included the precondition region of switch (shown in A.12). In total, we collected 60 responses but had to discard 12 due to the fact they reported not seeing the explanations. Table 1b presents the average time taken and steps taken to complete the task (along with 95% confidence intervals). As seen, the average value is much smaller for concept explanation. Additionally, a two-tailed t-test shows the results are statistically significant (p-values of 0.021 and 0.038 against a significance level $\alpha = 0.05$ for time and steps respectively).

## 8    CONCLUSION

We view the approaches introduced in the paper as the first step towards designing more general post hoc symbolic explanation methods for sequential decision-making problems. We facilitate the generation of explanations in user-specified terms for contrastive user queries. We implemented these methods in multiple domains and evaluated the effectiveness of the explanation using user studies and computational experiments. As discussed in (Kambhampati et al., 2021), one of the big advantages of creating such symbolic representation is that it provides a symbolic middle layer that can be used by the users to not only understand agent behavior, but also to shape it. Thus the user could use the model as a starting point to provide additional instructions or to clarify their preferences. Section A.11 contains more detailed discussion on various future works and related topics, including its applicability to settings with stochasticity, partial observability, temporally extended actions, the process of collecting more concepts, identifying confidence threshold and possible ethical implications of using our post-hoc explanation generation methods.

## 9 ACKNOWLEDGEMENTS

This research is supported in part by ONR grants N00014-16-1-2892, N00014-18-1-2442, N00014-18-1-2840, N00014-9-1-2119, AFOSR grant FA9550-18-1-0067, NSF 1909370, DARPA SAIL-ON grant W911NF19-2-0006 and a JP Morgan AI Faculty Research grant. We thank Been Kim, the reviewers and the members of the Yochan research group for helpful discussions and feedback.

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

## A APPENDIX

### A.1 OVERVIEW

This appendix contains the following information; (1) the assumptions and theoretical results related to the representational choices made by the method, (2) the pseudo-code for the probabilistic version of the algorithms which were used in the evaluation, (3) the derivation of the formulas for confidence calculation (4) the derivation of the formulas for using a noisy classifier and finally (5) the details on the experiment that are not included in the main paper (6) analysis of assumptions made and (7) extended discussion of various future works and possible limitations of the current method and (8) screenshots of the various interfaces.

### A.2 SUFFICIENCY OF THE REPRESENTATIONAL CHOICE

The central representational assumption we are making is that it is possible to approximate the applicability of actions and cost functions in terms of high-level concepts. Apart from the intuitive appeal of such models (many of these models have their origin in models from folk psychology), these representation schemes have been widely used to model real-world sequential decision-making problems from a variety of domains and have a clear real-world utility Benton et al. (2019).

Revisiting the definition of local approximations for a given internal model $\mathcal{M} = \langle S, A, T, \mathcal{C} \rangle$ and a set of states $\hat{S} \subseteq S$, we define a local approximation as

**Definition 3** *A symbolic model* $\mathcal{M}_{\hat{S}}^{\mathbb{C}} = \langle \mathbb{C}, A_{\hat{S}}^{\mathbb{C}}, \mathbb{C}(I), \mathbb{C}(\mathbb{G}), \mathcal{C}_{\hat{S}}^{\mathbb{C}} \rangle$. *is said to be a **local symbolic approximation** for the problem* $\Pi^R = \langle \mathcal{M}^R, I, \mathcal{G} \rangle$ *(where* $\mathcal{M}^R = \langle S, A, T, \mathcal{C} \rangle$*) for regions of interest* $\hat{S} \subseteq S$ *if* $\forall s \in \hat{S}$ *and* $\forall a \in A$, *we have an equivalent action* $a^{\mathbb{C}} \in A_{\hat{S}}^{\mathbb{C}}$, *such that (a)* $a^{\mathbb{C}}(\mathbb{C}(s)) = \mathbb{C}(T(s, a))$ *(assuming* $\mathbb{C}(\perp) = \perp$*) and (b)* $\mathcal{C}_{\hat{S}}^{\mathbb{C}}(\mathbb{C}(s), a) = \mathcal{C}(s, a)$ *and (c)* $\mathbb{C}(\mathbb{G}) = \bigcap_{s_g \in \mathbb{G} \cap \hat{S}} \mathbb{C}(s_g)$.

We can show that

**Proposition 1** *If* $\hat{S}$ *and* $A$ *are finite, there exists a symbolic model* $\mathcal{M}_{\hat{S}}^{\mathbb{C}} = \langle \mathbb{C}, A_{\hat{S}}^{\mathbb{C}}, \mathbb{C}(I), \mathbb{C}(\mathbb{G}), \mathcal{C}_{\hat{S}}^{\mathbb{C}} \rangle$ *such that it's a local symbolic approximation.*

We can show this trivially by construction. We introduce a set of concepts whose size is equal to the number of state in $\hat{S}$ (for $s_i \in \hat{S}$, we introduce a concept $c_{s_i}$). and we define a conditional effect and conditional cost function for every viable transition in for an action $a$ in $\mathcal{M}^R$, and the precondition of $a$ becomes a disjunction over the negation of concepts corresponding to the states where $a$ fails. In Proposition 3, we will further discuss how these disjunctive precondition turns into new conjunctive preconditions.

**Proposition 2** *If* $\pi_i$, $\pi_2$ *are two plans such that* $\pi_1 \preceq \pi_2$ *for* $\Pi^R$, *then precedence is conserved in a symbolic model* $\mathcal{M}_{\hat{S}}^{\mathbb{C}}$ *that is local approximation that covers all states appear in* $\pi_i$ *or* $\pi_2$.

This trivially follows from the fact that per definition of local approximation both the invalidity of plans and cost of plans are conserved. Which means any preference over plans that can be established in the complete model can be established in the approximate model.

Apart from this basic assumption, we make one additional representational assumption, namely, that the precondition can be expressed as a conjunction of positive concepts. Which brings us to the next proposition

**Proposition 3** *A precondition of an action* $prec_{a_i}$, *which is represented as an arbitrary logical formula over a set of propositions* $P$ *can be mapped to a conjunction over positive literals from a different set* $P'$ *(which can be generated from* $P$*).*

To see why this holds, consider a case where the precondition of action $a$ is expressed as an arbitrary propositional formula, $\phi(\mathbb{C})$. In this case, we can express it in its conjunctive normal form $\phi'(\mathbb{C})$. Now each clause in $\phi'(\mathbb{C})$ can be treated as a new compound positive concept. Thus we can cover

**a** Algorithm for Finding Missing Precondition

1: **procedure** PRECONDITION-SEARCH
2:     *Input*:  $s_{\text{fail}}, a_{\text{fail}}$, Sampler, $\mathcal{M}^R, \mathbb{C}, \ell, \kappa$
3:     *Output*: Missing precondition $C_{\text{prec}}$
4:     *Procedure*:
5:     $\mathbb{P} \leftarrow$ Missing concepts in $s_{fail}$
6:     $P_\mathbb{P} \leftarrow$ Initialize priors
7:     sample_count = 0
8:     **while** sample_count $< \ell$ **do**
9:         $s \sim$ Sampler
10:         **if** $T(s, a_{\text{fail}}) \neq \bot$ **then**
11:             Update $P_\mathbb{P}$
12:             Eliminate any $c_i \in \mathbb{P}$ , such that $P_\mathbb{P}(c_i) < \kappa$
13:         **if** $|\mathbb{P}| = 0$ **then return** Signal that concept list is incomplete
14:             sample_count += 1
        **return** $C_i \in$ poss_prec_set, with highest probability

**b** Algorithm for Finding Cost Function

1: **procedure** COST-FUNCTION-SEARCH
2:     *Input*:   $\pi_f, \mathcal{C}_\pi$, Sampler, $\mathcal{M}^R, \mathbb{C}, \ell$
3:     *Output*: $\mathbb{C}_{\pi_f}$
4:     *Procedure*:
5:     **for** conc_limit in 1 to $|\mathbb{C}|$ **do**
6:         current_foil_cost = 0
7:         Prob_list = []
8:         conc_list = []
9:         **for** i in 1 to k (the length of the foil) **do**
10:             $\hat{\mathbb{C}}_i$,    min_cost,    $P_{(\hat{\mathbb{C}}_i, a_i) \geq min\_cost}$ = find_min_conc_set($\mathbb{C}(T(s, \langle a_1, ...a_{i-1}\rangle)), a_i$, conc_limit, $\ell$)
11:             current_foil_cost += min_cost
12:             Prob_list.push($P_{(\hat{\mathbb{C}}_i, a_i) \geq min\_cost}$)
13:             conc_list.push($\hat{\mathbb{C}}_i$, min_cost)
14:         **if** current_foil_cost $> \mathcal{C}_\pi$ **then return** conc_list, Prob_list
    **return** Signal that the concept list is incomplete

Figure 6: The extension of the two algorithms to use the confidence values, Subfigure (a) presents the algorithm for missing precondition and (b) the one for cost function

such arbitrary propositional formulas by expanding our concept list with compound concepts (including negations and disjuncts) whose value is determined from the classifiers for the corresponding atomic concepts. Note that the set of all possible compound concepts could be precomputed and add no additional overhead on the human's end. Also note, this is completely compatible with relational concepts as relational concepts defined over a finite set of objects can be compiled into a set of finite propositional concepts. Proposition 3 also extends to cost functions, which we assume to be defined over conjunction of concepts.

### A.3   EXPLANATION IDENTIFICATION WITH PROBABILISTIC CONFIDENCE

Now the objective of the methods become not only identify an explanation but one that has the likelihood of being true. This means for precondition the objective becomes. Given the failing state $s_{\text{fail}}$ and action $a_{\text{fail}}$ and a set of states $\mathbb{S}$ where $a_{\text{fail}}$ is executable, explanation identification for a failing precondition involves finding a concept $c_i \in \mathbb{C} \setminus \mathbb{C}(s_{\text{fail}})$, such that $c_i = \arg\max_c P(c \in p_{a_{\text{fail}}}|\mathbb{S})$.

In terms of the algorithm for finding precondition, the main difference is the fact that instead of eliminating the hypothesis directly from their presence or absence in the given state, we will instead use the observation to update the posterior over the hypothesis being true. We can also further improve the efficiency of search by removing possible hypothesis for final precondition, when its probability dips below a low threshold $\kappa$. Note that when $\kappa = 0$ and the observation model is perfect ($P(O^s_{c_i}|c_i \in s) = 1$ and $P(O^s_{c_i}|c_i \not\in s) = 0$), then the reasoning reflects the elimination based model learning method used by many of the previous approaches like Carbonell & Gil (1990); Stern & Juba (2017).

Now for the cost function, the objective becomes

**Definition 4** *Given a foil $\pi^f = \langle a_1, ..., a_f \rangle$ with $m \leq f$ unique actions (where $A(a_i)$ returns the unique action label), a set sets of states $\{\mathbb{S}_1, ..., \mathbb{S}_m\}$ where each $\mathbb{S}_{A(a_i)}$ corresponds to a set of states where an action $A(a_i)$ is executable, the explanation for suboptimality correpond to finding an abstract cost function, that tries to minimize for the cost and maximize the probabilities.*

$$min_{\hat{\mathbb{C}}_1, ..., \hat{\mathbb{C}}_f}(\sum_{i=1..f} \|\hat{\mathbb{C}}_i\|, -1 \times P(\mathcal{C}(\hat{\mathbb{C}}_1, a_1) \geq k_1|\mathbb{S}_{A(a_1)}), ..., -1 \times P(\mathcal{C}(\hat{\mathbb{C}}_f, a_f) \geq k_f|\mathbb{S}_{A(a_k)}))$$

$$\text{subject to } \mathcal{C}^{abs}_s(\mathbb{C}_{\pi_f}, \pi_f) > \mathbb{C}(I, \pi)$$

Rather than solve this full multi-objective optimization problem, we will use the cost as the primary optimization criteria and use probabilities as a secondary one. Figure 6(b), present a modified version of the greedy algorithm to find such cost abstractions. The procedure find_min_conc_set, takes

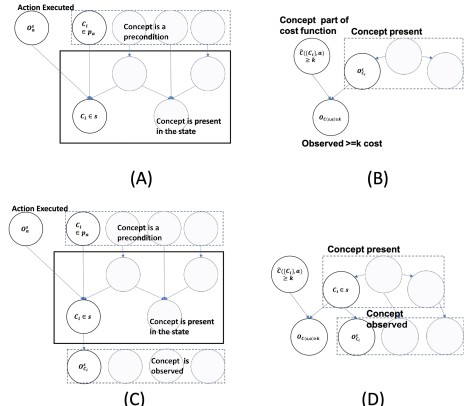

Figure 7: A simplified probabilistic graphical models for explanation inference, Subfigure (A) and (B) assumes classifiers to be completely correct, while (C) and (D) presents cases where the classifier may be noisy.

the current concept representation of state $i$ in the foil and searches for the subset $\hat{\mathbb{C}}_i$ (and its probabilistic confidence) of the state with the maximum value for $\mathcal{C}_{\mathcal{S}}^{abs}(\hat{\mathbb{C}}_i, a_i)$, where the value is again approximated through sampling (with budget $\ell$), and the subset size is upperbounded by conc_limit. If there are multiple abstract cost function with the same max cost it selects the one with the highest probability. Similar to the algorithm described in the main paper, here we incrementally increase the max concept size till we find an abstract cost function that meets the requirement.

## A.4 Confidence Calculation

For confidence calculation, we will be relying on the relationship between the random variables as captured by Figure 7 (A) for precondition identification and Figure 7 (B) for cost calculation. Where the various random variables captures the following facts: $O_a^s$ - indicates that action $a$ can be executed in state $s$, $c_i \in p_a$ - concept $c_i$ is a precondition of $a$, $O_{c_i}^s$ - the concept $c_i$ is present in state $s$, $\mathcal{C}_s^{abs}(\{c_i\}, a) \geq k$ - the abstract cost function is guaranteed to be higher than or equal to k and finally $O_{\mathcal{C}(s,a)>k}$ - stands for the fact that the action execution in the state resulted in cost higher than or equal to $k$. Since we don't know the exact relationship between the current concept in question is, for notational convenience we will use the symbol $O_{\mathbb{C}(s)\setminus c_i}$ to stand for the other concepts observed in the given state.

We will allow for inference over these models, by relying on the following simplifying assumptions - (1) the distribution of all non-precondition concepts in states where the action is executable is the same as their overall distribution across the problem states (which can be empirically estimated), (2) cost distribution of an action over states corresponding to a concept that does not affect the cost function is identical to the overall distribution of cost for the action (which can again be empirically estimated). The first assumption implies that the likelihood of seeing a non-precondition concept in a sampled state is equal to the likelihood of it appearing in any sampled state. In the most general case this distribution can be given as $P(c_i|O_{\mathbb{C}(s)\setminus c_i})$, i.e. the likelihood of seeing this concept given the other concepts in the state. This distribution can be empirically estimated per user (or shared vocabulary) independent of the specific explanatory query. While the second one implies that for a concept that has no bearing on the cost function for an action, the likelihood that executing the action in a state where the concept is present will result in a cost greater than $k$ will be the same as that of the action execution resulting in a cost greater than $k$ for a randomly sampled state ($p_{\mathcal{C}(\cdot,a)\geq k}$). This assumption can be further relaxed by considering the distribution of the action cost under the observed set of concepts (though this would clearly require more samples to learn).

For a single sample, the posterior probability of explanations for each case can be expressed as follows: For precondition estimation, updated posterior probability for a positive observation can be computed as $P(c_i \in p_a|O_{c_i}^s \wedge O_a^s \wedge O_{\mathbb{C}(s)\setminus c_i}) = (1 - P(c_i \notin p_a|O_{c_i}^s \wedge O_a^s \wedge O_{\mathbb{C}(s)\setminus c_i}))$, where

$$P(c_i \notin p_a | O^s_{c_i} \wedge O^s_a \wedge O_{\mathbb{C}(s)\backslash c_i})$$
$$= \frac{P(O^s_{c_i} | c_i \notin p_a \wedge O^s_a \wedge O_{\mathbb{C}(s)\backslash c_i}) * P(c_i \notin p_a | O^s_a \wedge O_{\mathbb{C}(s)\backslash c_i})}{P(O^s_{c_i} | O^s_a \wedge O_{\mathbb{C}(s)\backslash c_i})}$$

Given $c_i \notin p_a$ is independent of $O^s_a$ and $O_{\mathbb{C}(s)\backslash c_i}$ also expanding the denominator we get

$$= \frac{P(O^s_{c_i} | c_i \notin p_a \wedge O^s_a \wedge O_{\mathbb{C}(s)\backslash c_i}) * P(c_i \notin p_a)}{\begin{array}{c} P(O^s_{c_i} | c_i \notin p_a \wedge O^s_a \wedge O_{\mathbb{C}(s)\backslash c_i}) * P(c_i \notin p_a) + \\ P(O^s_{c_i} | c_i \in p_a \wedge O^s_a \wedge O_{\mathbb{C}(s)\backslash c_i}) * P(c_i \in p_a) \end{array}}$$

From our assumption, we know $P(O^s_{c_i} | c_i \notin p_a \wedge O^s_a \wedge O_{\mathbb{C}(s)\backslash c_i})$ is same as the distribution $c_i$ over the problem states $(p(c_i | O_{\mathbb{C}(s)\backslash c_i}))$ and $P(O^s_{c_i} | c_i \in p_a \wedge O^s_a)$ must be one.

$$= \frac{p(c_i | O_{\mathbb{C}(s)\backslash c_i}) * P(c_i \notin p_a)}{p(c_i | O_{\mathbb{C}(s)\backslash c_i}) * P(c_i \notin p_a) + P(c_i \in p_a)}$$

For cost calculation, we can ignore $O_{\mathbb{C}(s)\backslash c_i}$, since according to the graphical model once the concept $c_i$ is observed, $O_{\mathcal{C}(s,a)\geq k}$ is independent of the other concepts.

$$P(\mathcal{C}^{abs}_s(\{c_i\},a) \geq k | O^s_{c_i} \wedge O_{\mathcal{C}(s,a)\geq k}) = \frac{P(O_{\mathcal{C}(s,a)\geq k} | O^s_{c_i} \wedge \mathcal{C}^{abs}_s(\{c_i\},a) \geq k) * P(\mathcal{C}^{abs}_s(\{c_i\},a) \geq k | O^s_{c_i})}{P(O_{\mathcal{C}(s,a)\geq k} | O^s_{c_i})}$$

Where $P(O_{\mathcal{C}(s,a)\geq k} | O^s_{c_i}, \mathcal{C}^{abs}_s(\{c_i\},a) \geq k)$ should be 1 and $\mathcal{C}^{abs}_s(\{c_i\},a) \geq k$ independent of $O^s_{c_i}$. Which gives

$$= \frac{P(\mathcal{C}^{abs}_s(\{c_i\},a) \geq k)}{P(O_{\mathcal{C}(s,a)\geq k} | O^s_{c_i})}$$

$$= \frac{P(\mathcal{C}^{abs}_s(\{c_i\},a) \geq k)}{\begin{array}{c} P(O_{\mathcal{C}(s,a)\geq k} | O^s_{c_i}, \mathcal{C}^{abs}_s(\{c_i\},a) \geq k)) * P(\mathcal{C}^{abs}_s(\{c_i\},a) \geq k)) + \\ P(O_{\mathcal{C}(s,a)\geq k} | O^s_{c_i} \wedge \neg\mathcal{C}^{abs}_s(\{c_i\},a) \geq k)) \times P(\neg\mathcal{C}^{abs}_s(\{c_i\},a) \geq k)) \end{array}}$$

From our assumptions, we have $P(O_{\mathcal{C}(s,a)\geq k} | O^s_{c_i} \wedge \neg\mathcal{C}^{abs}_s(\{c_i\},a) \geq k)) = p_{\mathcal{C}(.,a)\geq k}$

$$= \frac{P(\mathcal{C}^{abs}_s(\{c_i\},a) \geq k)}{P(\mathcal{C}^{abs}_s(\{c_i\},a) \geq k)) + p_{\mathcal{C}(.,a)\geq k} * P(\neg\mathcal{C}^{abs}_s(\{c_i\},a) \geq k))}$$

### A.5 Using Noisy Concept Classifiers

Note that in previous sections, we made no distinction between the concept being part of the state and actually observing the concept. Now we will differentiate between the classifier saying that a concept is present ($O^s_{c_i}$) is a state from the fact that the concept is part of the state ($c_i \in \mathbb{C}(S)$). Here we note that the learned relationship is over the actual concepts in the state rather than the observation (and thus we would need to learn it from states with true concept labels). The relationship between the random variables can be found in Figure 7 (C) and (D). We will assume that the probability of the classifier returning the concept being present is given by the probabilistic confidence provided by the classifier. Of course, this still assumes the classifier's model of its prediction is accurate. However, since it is the only measure we have access to, we will treat it as being correct. Now we can use this updated model for calculating the confidence. For the precondition estimation, we can update the posterior of a concept being a precondition given a negative observation ($O^s_{\neg c_i}$) using the formula

$$P(c_i \notin p_a | O^s_{\neg c_i} \wedge O^s_a \wedge O_{\mathbb{C}(s)\backslash c_i}) = \frac{P(O^s_{\neg c_i} | c_i \notin p_a \wedge O^s_a \wedge O_{\mathbb{C}(s)\backslash c_i}) * P(c_i \notin p_a | O^s_a \wedge O_{\mathbb{C}(s)\backslash c_i})}{P(O^s_{\neg c_i} | O^s_a \wedge O_{\mathbb{C}(s)\backslash c_i})}$$

Where $P(c_i \notin p_a | O^s_a \wedge O_{\mathbb{C}(s)\backslash c_i}) = P(c_i \notin p_a)$ and we can expand $P(O^s_{\neg C_i} | c_i \notin p_a \wedge O^s_a \wedge O_{\mathbb{C}(s)\backslash c_i})$ as follows

$$P(O_{\neg c_i} | c_i \notin p_a \wedge O^s_a \wedge O_{\mathbb{C}(s)\backslash c_i}) =$$
$$P(O_{\neg c_i} | c_i \in \mathbb{C}(s)) * P(C_i \in \mathbb{C}(s) | c_i \notin p_a \wedge O^s_a \wedge O_{\mathbb{C}(s)\backslash c_i}) +$$
$$P(O_{\neg c_i} | c_i \notin \mathbb{C}(s)) * P(c_i \notin \mathbb{C}(s) | c_i \notin p_a \wedge O^s_a \wedge O_{\mathbb{C}(s)\backslash c_i})$$

Where as defined earlier $P(c_i \notin \mathbb{C}(s)|c_i \notin p_a \wedge O_a^s \wedge O_{\mathbb{C}(s)\backslash c_i})$ and $P(c_i \in \mathbb{C}(s)|C_i \notin p_a \wedge O_a^s \wedge O_{\mathbb{C}(s)\backslash c_i})$ would get expanded to the learned relationship between the concepts and their corresponding observation model. The denominator also needs to be marginalized over $c_i \notin \mathbb{C}(s)$.

Similarly for posterior calculation for positive observations, we have

$$P(O_{c_i}^s|c_i \notin p_a \wedge O_a^s \wedge O_{\mathbb{C}(s)\backslash c_i}) =$$
$$P(O_{c_i}|c_i \in \mathbb{C}(s)) * P(c_i \in \mathbb{C}(s)|c_i \notin p_a \wedge O_a^s \wedge O_{\mathbb{C}(s)\backslash c_i}) +$$
$$P(O_{c_i}|c_i \notin \mathbb{C}(s)) * P(c_i \notin \mathbb{C}(s)|c_i \notin p_a \wedge O_a^s \wedge O_{\mathbb{C}(s)\backslash c_i})$$

Now for the cost, we can similarly incorporate the observation model as follows.

$$P(\mathcal{C}_s^{abs}(\{c_i\}, a) \geq k|O_{c_i}^s \wedge O_{\mathcal{C}(s,a)>=k}) = \frac{\begin{aligned}P(O_{\mathcal{C}(s,a)\geq k}, O_{c_i}^s|\mathcal{C}_s^{abs}(\{c_i\}, a) \geq k)\\ * P(\mathcal{C}_s^{abs}(\{c_i\}, a) \geq k)\end{aligned}}{P(O_{\mathcal{C}(s,a)\geq k}, O_{c_i}^s)}$$

$$= \frac{\begin{aligned}(P(O_{\mathcal{C}(s,a)\geq k}, O_{c_i}^s|c_i \in \mathbb{C}(s), \mathcal{C}_s^{abs}(\{c_i\}, a) \geq k) * P(c_i \in \mathbb{C}(s)) +\\ P(O_{\mathcal{C}(s,a)>k}, O_{c_i}^s|c_i \notin \mathbb{C}(s), \mathcal{C}_s^{abs}(\{c_i\}, a) \geq k) * P(c_i \notin \mathbb{C}(s))) * P(\mathcal{C}_s^{abs}(\{c_i\}, a) \geq k)\end{aligned}}{P(O_{\mathcal{C}(s,a)\geq k}, O_{c_i}^s)}$$

Given their parents, $O_{\mathcal{C}(s,a)\geq k}$ and $O_{c_i}^s$ are conditionally independent, and given its parent $O_{c_i}^s$ is independent of $\mathcal{C}_s^{abs}(\{c_i\}, a) \geq k)$, there by giving

$$= \frac{\begin{aligned}(P(O_{\mathcal{C}(s,a)\geq k}|c_i \in \mathbb{C}(s), \mathcal{C}_s^{abs}(\{c_i\}, a) \geq k) * P(O_{c_i}^s|c_i \in \mathbb{C}(s)) * P(c_i \in \mathbb{C}(s)) +\\ P(O_{\mathcal{C}(s,a)\geq k}|c_i \notin \mathbb{C}(s), \mathcal{C}_s^{abs}(\{c_i\}, a) \geq k) * P(O_{c_i}^s|c_i \notin \mathbb{C}(s)) * P(c_i \notin \mathbb{C}(s))) * P(\mathcal{C}_s^{abs}(\{c_i\}, a) \geq k)\end{aligned}}{P(O_{\mathcal{C}(s,a)\geq k}, O_{c_i}^s)}$$

Now $P(O_{\mathcal{C}(s,a)\geq k}|c_i \in \mathbb{C}(s), \mathcal{C}_s^{abs}(\{c_i\}, a) \geq k) = 1$ and $P(O_{\mathcal{C}(s,a)\geq k}|c_i \notin \mathbb{C}(s), \mathcal{C}_s^{abs}(\{c_i\}, a) \geq k)$ can either be empirically estimated from true labels or we can make the assumption that is equal to $p_{\mathcal{C}(.,a)\geq k}$ (which we made use of in our experiments), which would take us to

$$= \frac{\begin{aligned}(P(O_{c_i}^s|c_i \in \mathbb{C}(s))P(c_i \in \mathbb{C}(s)) +\\ p_{\mathcal{C}(.,a)\geq k} * P(O_{c_i}^s|c_i \notin \mathbb{C}(s)) * P(c_i \notin \mathbb{C}(s))) * P(\mathcal{C}_s^{abs}(\{c_i\}, a) \geq k)\end{aligned}}{P(O_{\mathcal{C}(s,a)\geq k}, O_{c_i}^s)}$$

## A.6 EXPERIMENT DOMAINS

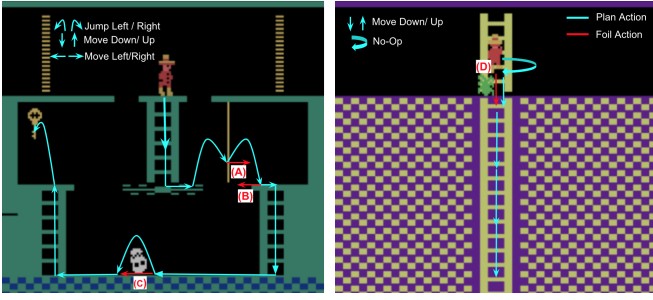

Figure 8: Montezuma Foils: Left Image shows foils for level 1, (A) Move right instead of Jump Right (B) Go left over the edge instead of using ladder (C) Go left instead of jumping over the skull. Right Image shows foil for level 4, (D) Move Down instead of waiting.

For validating the soundness of the methods discussed before, we tested the approach on the open-AI gym implementation of Montezuma's Revenge Brockman et al. (2016) and variants of Sokoban Schrader (2018). Most of the search experiments were run on an Ubuntu 14.0.4 machine with 64 GB RAM.

For Montezuma, we used the deterministic version of the game with the RAM-based state representation (the game state is represented by the RAM value of the game controller, represented by

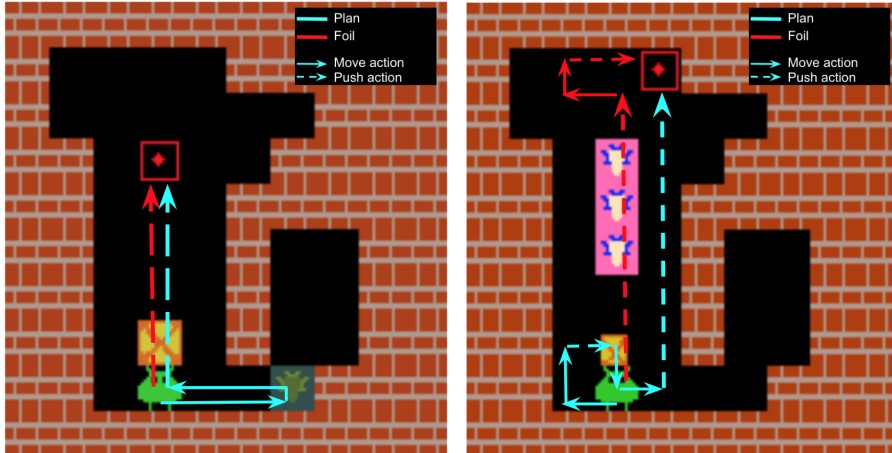

Figure 9: Sokoban Foils: Left Image shows foils for Sokoban-switch, note that the green cell will turn pink once the agent passes it. Right Image shows foil for Sokoban-cell.

256-byte array). We considered executing an action in the simulator that leads to the agent's death (falling down a ledge, running into an enemy) or a non-NOOP (NOOP action is a specific agent action that is designed to leave agent's state unchanged) action that doesn't alter the agent position (trying to move left on a ladder) as action failures. We selected four possible foils for the game (illustrated in Figure 8), three from level 1 and one from level 4. The base plan in level 1 involves the agent reaching the key, while level 4 required the agent to cross the level.

For Sokoban, we considered two variants with a single box and a single target. Both variants allow for 8 actions and a NOOP action. Four of those actions are related to the agent's movements in the four directions and four to pushing in specific directions. We restricted the move actions only to be able to move the agent if the cell is empty, i.e., it won't move if there is a box or a wall in that direction. The push action also moves the agent in the direction of push if there is a box in that direction, and the agent will occupy the cell previously occupied by the box (provided there are no walls to prevent the box from moving). Similar to Montezuma, we consider any action that doesn't change the agent position to be a failure. The first version of Sokoban included a switch the player could turn on to push the box (we will refer to this version as Sokoban-switch), and the second version (Sokoban-cells) included particular cells from which it is costlier to push the box. We considered two versions of Sokoban-switch, one in which turning on the switch only affected the cost of pushing the box and another one in which it was a precondition. For the cost case of Sokoban-switch, while the switch is on (i.e. the cell is pink), all actions have unit cost, while when the switch is off, the cost of pushing actions is 10. The cell can be switched on by visiting it, and any future visit will cause it to turn off. For the precondition-version, pushing of the box while the switch is off causes the episode to end with high cost (100). Since we also trained an RL agent for this version for generating the saliency map. We also added a small penalty for not pushing the switch and a penalty proportional to the distance between the box and the target. In Sokoban-cells, the cost of all actions except pushing boxes in the pink region is one, while that of pushing boxes in the pink region is 10. We selected one foil per variation, and the original plan and the foil are shown in Figure 9.

**Concept Learning** For Montezuma, we came up with ten base concepts for each level that approximates the problem dynamics at the level, including the foil failure. We additionally created ten more concepts by considering the negations of them. All state samples (used to generate the samples for the classifier and the algorithm) were created by randomly selecting one of the states from the original plan and then performing random walks from the selected state. For the classifiers, we used game-specific logic and RAM byte values to identify each positive instance and then randomly selected a set of negative examples. We used around 600 positive examples (except for the concepts $skull\_on\_right$ and $skull\_on\_left$ in level 1, which had 563 and 546 examples, respectively) and twice as many negative examples for each concept. These RAM state examples were

fed to a binary AdaBoost Classifier Freund et al. (1999) (Scikit-learn implementation Pedregosa et al. (2011) version 0.22.1, with default parameters), with 70% of samples used as train set and the rest as the test set, for each concept. Finally, we obtained a test accuracy range of 98.57% to 100%, with an average of 99.72% overall concepts of both the levels. All the samples used for the classifier were collected from 5000 sampling episodes for level 1 and 4000 sampling episodes for level 4. During the search, We used a threshold of 0.55 on classifiers for concepts of level 1, such that a given state has a given concept when the classifier probability is greater than 0.55, to reduce false positives. The code for sampling and training the classifiers can be found in the directory PRECOND_BLACKBOX/sampler_and_conceptTrain inside the code directory.

### A.7 CONCEPT COLLECTION

For the Sokoban variants, we wanted to collect at least the list of concepts from people. We used surveys to collect the set of concepts. The survey allowed participants to interact with the game through a web interface, and at the end, they were asked to specify game concepts that they thought were relevant for particular actions. Each user was asked to specify a set of concepts that they thought were relevant for four actions in the game. They were introduced to the idea of concepts and their effect on the action by using PACMAN as an example and presenting three example concepts. The exact instructions and screenshots of the interface used for Sokoban-cell can be found in the file Sokoban_cell_survey.pdf in the directory Study_Website_Pdfs, which is part of the supplementary file zip. For Sokoban-switch, we collected data from six participants, four of whom were asked to specify concepts for push actions and two people for move actions. For Sokoban-cell, we collected data from seven participants, six of whom were asked to specify concepts for push actions, and one was asked to specify concepts for move action. We went through the submitted concepts and clustered them into unique concepts using their description. We skipped ones where they just listed strategies rather than concepts describing the state. We removed two concepts from the Sokoban-cell list and two from Sokoban-switch because we couldn't make sense of the concept being described there. For Sokoban-switch, we received 25 unique concepts, and for Sokoban-cell, we collected 38 unique concepts. We wrote scripts for each of the concepts and used it to sample example states. We ran the sampler for 1000 episodes to collect the examples for the concepts. We trained classifiers for each of the concepts that generated more than 10 positive examples for the concepts. For sokoban-switch, we removed two additional concepts because their training set didn't contain any positive examples. We had, on average, 178.46 positive examples for Sokoban-cell per concept and 215.55 for Sokoban-switch. We used all the other samples as negative examples. We again used 70% of samples for training and the remaining for testing. We used Convolutional Neural Networks (CNNs) based classifiers for the Sokoban variants. The CNN architecture involved four convolutional layers, followed by three fully connected layers that give a binary classification output. The average accuracy of the Sokoban-switch was 99.46%, and Sokoban-cell was 99.34%. The code used for sampling and training for each domain can be found under the folder COST_TRAINER (inside the directory BLACKBOX_CODE). The classifier network is specified in the file CNNnetwork.py. The details on how to run them are provided in the README file in the root code directory.

### A.8 EXPLANATION IDENTIFICATION

For Montezuma, the concept distribution was generated using 4000 episodes, and the probability distribution of concepts ranged from 0.0005 to 0.206. For some of the less accurate models, we did observe false negatives resulting in the elimination of the accurate preconditions and empty possible precondition set. So we made use of the probabilistic version of the search with observation probabilities calculated from the test set. We applied a concept cutoff probability of 0.01, and in all cases, the precondition set reduced to one element (which was the expected precondition) in under the 500 step sampling budget (with the mean probability of 0.5044 for foils A, B & C. Foil D, in level 4, gave a confidence value of 0.8604). The ones in level 1 had lower probabilities since they were based on more common concepts, and thus, their presence in the executable states was not strong evidence for them being a precondition.

For each Sokoban variant, we ran another 1000 episode sampler, which used random restarts from the foil states to collect samples that we used for the explanation generation. During the generation of the samples, we used the previously learned classifiers to precompute the mapping from concepts to states.

We used a variant of Figure 6(b), where we sped up the search by allowing for memoization. Specifically, when sampling is done for an action and a specific conc_limit for the first time, then we precompute the min-cost for all possible concept subset of that size. Then for every step that uses that action, we look up the min value for the subset that appears in the state. The search was run with a sampling budget of 750. For calculating the confidence, all required distributions were calculated only on the states where the action was executable. Again the search was able to find the expected explanation. We had average confidence of 0.9996 for the Sokoban-switch and 0.998 for the Sokoban-cell. The exact observation models values used can be found in constant.py under the directory COST_BLACKBOX/src, and the file cost_explainer.py in the same directory contains the code for the exact search we used.

In terms of time taken to generate the sample, creating 100 samples from montezuma for the first screen (averaged across the foils) took 59.754 seconds, montezuma second screen it took 97.673 seconds (the additional time is due to the overhead of the agent being moved to the new level before the start of the episode) and the Sokoban variants took 40.660 secs.

## A.9  USER STUDY

With the basic explanation generation method in place, we were interested in evaluating if users would find such an explanation helpful. Specifically, the hypotheses we tested were

**H1**: *People would prefer explanations that establish the corresponding model component over ones that directly presents the foil information (i.e. the failing action and per-step cost)*

**H2**: *Concept-based precondition explanations help users understand the task better than saliency map based ones.*

To evaluate this, we performed a user study with all the foils used along with the generated explanation and a simple baseline. In the study, each participant was presented with a random subset of the concept we used for the study (around five) and then was shown the plan and a possible foil. Then the participant was shown two possible explanations for the foil (the generated one and the baseline) and asked to choose between them. There were additional questions at the end asking them to specify what they believed was the completeness of the selected explanation, on a Likert scale from 1 to 5 (1 being least complete and 5 being the most). They were also provided a free text field to provide any additional information they felt would be useful.

For precondition explanation, the options showed to the user includes one that showed the state at which the foil failed along with the information that the action cannot be executed in that state and the other one reported that the action failed because a specific concept was missing (the order of the options was randomized). In total, we collected data from 20 participants, where 7 were women, the average age was 25, and 10 people had taken an AI class. We found that 19 out of the 20 participants selected precondition based explanation as a choice. On the question of whether the explanation was complete, we had an average score of 3.476 out of 5 on the Likert scale (1 being not at all complete and 5 being fully complete). The results seem to suggest that information about missing precondition are useful explanations though these may not be complete. While not a lot of participants provided information on what other information would have been useful, the few examples we had generally pointed to providing more information about the model (for example, information like what actions would have been possible in the failed state).

For the cost function explanation, the baseline involved pointed out the exact cost of executing each action in the foil, and concept-explanation showed how certain concepts affected the action costs. In the second case, all action costs were expressed using the abstract cost function semantics in that they were expressed as the 'action costing at least X' even though in our case, the cost was the same as the abstract cost. For the cost condition, again, we collected 20 replies in total (ten per foil) and found 14 out of 20 participants selected the concept-based explanation over the simple one. The concept explanations had, on average, a completeness score of 3.214 out of 5. The average age of the participants was 24.15, 10 had AI knowledge out of 20 people, 11 were masters students (rest were undergrad), and we had 14 males, 5 females, and one participant did not specify.

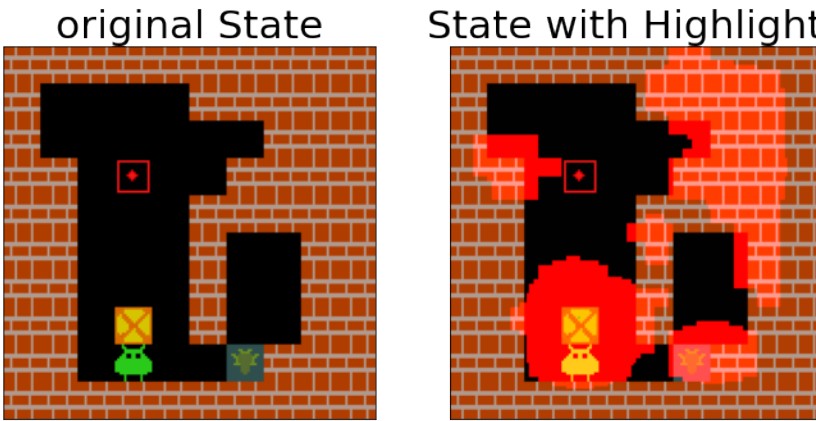

Figure 10: Saliency map based explanation shown to users as part of H2

For H2, as mentioned the baseline was a saliency map based explanation. For generating the saliency map, we trained the RL agent using DQN with prioritized experience replay Schaul et al. (2015)[2]. The agent was trained for 420k epochs. The Saliency map itself was generated for four states, with the agent placed on the four sides of the box. The saliency map itself was generated using Greydanus et al. (2018), where we used only the procedure for generating the map for the critic network[3]. Figure 10 shows the saliency map generated for one of the images. This was shown when the user tried push up action and fails. At the beginning of the study, both groups of the users were made to familiarize themselves with five concepts that were randomly ordered (the concepts themselves remained the same) and had to take a quiz matching new states to those concepts, before moving on to play the game. Out of the 60 responses we considered, 16 identified as female and 41 identified as men. 23 of the participants reported they had some previous knowledge of AI, but only three participant reported having any planning knowledge. *The participants who got concept based explanations took 43.78 ± 12.59 secs (95% percentile confidence) and 35.87 ± 9.69 steps on average to complete the game. On the otherhand participants of the other group took 134.24 ± 61.72 secs and 52.64 ± 11.11 steps on average.*

PDF files showing the screenshots of the user study website for a scenario for the precondition explanation test and one from cost explanation can be found in User-study-precondition.pdf and User-study-cost.pdf in the directory Study_Website_Pdfs. The actual data collected from the user studies and the concept survey can be found in the directory USER_STUDY_FEEDBACK. But below, we have included some screenshots of the interface

## A.10    ANALYSIS OF ASSUMPTIONS MADE FOR CONFIDENCE CALCULATIONS

In this section we present results from additional tests we ran to verify some of the assumptions made in the confidence calculations on the Sokoban variants. We mainly focused on Sokoban variants since the concepts were collected directly from users and we tested the assumptions - (1) the distribution of all non-precondition concepts in states where the action is executable is the same as their overall distribution across the problem states (which can be empirically estimated) and (2) cost distribution of an action over states corresponding to a concept that does not affect the cost function is identical to the overall distribution of cost for the action (which can again be empirically estimated). We didn't run a separate test on the independence of concepts as we saw that many of the concepts listed by the users were in fact correlated and were denoting similar or even the same phenomena. All concepts were assumed to be distributed according to a Bernoulli distribution, whose MLE estimates were calculated by running the sampler for ten thousand episode, where we used states from the original successful/optimal plan as the initial state for the random walk (ensuring the

---

[2]For exact agent, we followed the approach described in `https://github.com/higgsfield/RL-Adventure/blob/master/4.prioritized%20dqn.ipynb`

[3]We made use of the code available at `https://github.com/greydanus/visualize_atari` which had a GPL licence

| Action | Concept with Max difference | Max Absolute Difference in Estimates | Average Difference in Estimates |
|---|---|---|---|
| push up | empty_above | 0.018 | 0.004 |
| push down | empty_below | 0.0154 | 0.0033 |
| push left | empty_below | 0.0163 | 0.0037 |
| push right | empty_below | 0.0172 | 0.0046 |
| move up | empty_left | 0.002 | 0.0009 |
| move down | wall_left | 0.0017 | 0.0007 |
| move left | empty_right | 0.0032 | 0.0009 |
| move right | empty_right | 0.0045 | 0.0008 |

Figure 11: Results from Sokoban-switch on the distribution of non-precondition concepts for each action. For each action the table reports the concept with the maximum difference between the distribution of concept for that specific version, versus the overall distribution and the average difference in estimates across concepts.

| Domain | Action | Concept with Max difference | Max Absolute Difference in Estimates | Average Difference in Estimates |
|---|---|---|---|---|
| Sokoban-Switch | push up | wall_left_below_of_box | 0.1132 | 0.0461 |
| | push down | wall_left_below_of_box | 0.1107 | 0.0473 |
| | push left | above_switch | 0.1135 | 0.0461 |
| | push right | wall_left_below_of_box | 0.111 | 0.0479 |
| Sokoban-Cell | push up | box_on_right | 0.0956 | 0.0411 |
| | push down | box_on_right | 0.1098 | 0.0476 |
| | push left | box_on_right | 0.1012 | 0.0486 |
| | push right | wall_on_left | 0.0889 | 0.0433 |

Figure 12: Results from Sokoban-switch and Sokoban-Cell on the distribution of action cost across different concepts. Here we report only the cost for push actions, since only those actions result in higher cost.

distributions are generated from state space local to the plan of interest). For first assumption, we compared the distribution of concept for states where the action was executed against the distribution of the concept over all the sampled states. For the second assumption, we compared the distribution of the states with the high cost ($>=10$) where the concept is present versus the distribution of high cost for the action.

Table 11, summarizes the results from testing the first assumption for Sokoban-switch. For each action the table reports the non-precondition concept which had the maximum difference in estimates (the reported difference in the table). In this domain, the only precondition concept is switch_on for the push actions. As we can see for the domain, the differences are pretty small and we expect the differences to further reduce once we start accounting the correlation between concepts.

Table 12 presents the results for the second assumption. In the cases of Sokoban-switch, we again skipped the switch_on concept and for Sokoban-Cell we skipped the concepts related to the pink cells since they are all highly correlated to central concept controlling the cost function (on_pink_cell).

## A.11 EXTENDED DISCUSSION

**Collecting Concepts** In most of the current text, we expect the set of concepts and classifiers to be given. As such before the system is actually used in practice we would have a stage where the initial set of concepts are collected. Collecting all the concepts from the same user may be taxing, even if we made use of more straightforward methods to learn the classifiers. Instead, a more scalable method may be to set up domain-specific databases for each task that includes the set of commonly used concepts for the task. This was also the strategy used by Cai et al. (2019), who created a medical concept database to be used along with TCAV explanations. One could also use off-the-shelf object recognition and scene graph generation methods to identify concepts for everyday settings.

**Describing Plans:**   While contrastive explanations are answers to questions of the form "Why P and not Q?", we have mostly focused on refuting the foil (the "not Q?" part). For the "Why P" part the agent could start by demonstrating its plan to show why it is valid and report the total cost it may accumulate. We can further augment such traces with the various concepts that are valid at each step of the trace. We can also report a cost abstraction for the cost function corresponding to each step taken as part of the plan. Here the process for finding the cost abstraction stays the same as what is described in Section 4 of the main paper.

**Stochastic Domains:**   While most of the discussions in the paper have been focused on deterministic domains, *these ideas also carry over to stochastic domains*. The way we are identifying preconditions and estimating cost functions remain the same in stochastic domains. The only difference would be the estimation of the value or the failure point of the foil. One way to easily adapt them to our method would be to compare against the worst case, or best case execution cost of the foil or the failure point under one of the execution traces corresponding to the foil. So the only change we would need to make to the entire method is before the actual search starts. Namely, when the agent is trying to simulate the foil in its own model, rather than doing it once, it would need to simulate it multiple times and look for the worst-case or best-case trace. Another possibility is rather than looking for the worst-case execution trace, would be to consider the most likely traces and perform the same process but now over a set of possible foils.

**Partial Observability and Non-Markovian Concepts:**   The paper focuses on cases where the mapping is from each state to concept. Though there may be cases where the concept the user is interested in corresponds to a specific trajectory. Or there may be cases where there may be some disparity between the internal state of the agent and what the human can observe, or the problem itself is partially observable. In such cases, rather than learning a concept that is completely identified by the current state, the system may need to learn mappings from state action trajectories (or observation histories) to concepts. Then the rest of the process stays the same (except the search process now needs to keep track of history). We can still use the same class of symbolic models as before because while these concepts are non-markovian with regards to the underlying model, the fact that the concept symbols are part of the symbolic model state means any action transition or cost function that depends on these concepts can be defined purely with respect to the current state.

**Temporally Extended Actions:**   Another assumption we have made through the paper is that the action space stays the same across the symbolic and internal models. However, this may not always be true. For domains like robotics, the human may be reasoning about the actions at a higher level of abstractions than what the agent may be considering. Such actions may be modeled as temporal abstractions (for example, options Sutton et al. (1999)) defined over the agent's atomic actions. If such abstract actions are not pre-specified to the agent, it could try to learn it by a process similar to how it learned concepts. Once learned, the agent can either directly simulate these temporally abstract actions to identify its preconditions (for options, a characterization of its initiation set) and cost function (the expected cost over the holding time) or from the sampled traces which correspond to the execution of these abstract actions (if the system could only learn a mapping from trajectories to action labels).

**Acquiring New Concepts:**   In the main text, we discussed how the original vocabulary set may be incomplete and how our search algorithms could identify scenarios when the concept list is incomplete and insufficient to generate the required explanation. In such cases, the system would need to gather more concepts from the user. While the system could ask the user for any previously unspecified concepts, this would just result in the user wasting time specifying potentially irrelevant concepts. One would preferably want to perform a more directed concept acquisition. One possibility might be to use the system's internal model to provide hints to the user as to what concepts may be useful. For example, the system could generate low-level explanations like saliency maps to highlight the parts of the state that may be important and ask for concepts that are relevant to those states. Once the new concepts are collected, we can again make use of our current approach to see if any of the concepts could be used to build a satisfactory explanation. One could also wrap the entire interaction into a meta-sequential decision-making framework, where the framework captures the repeated interaction between the user and the system. The actions that are available to the meta decision-maker would include the ability to query the user for more concepts and the objec-

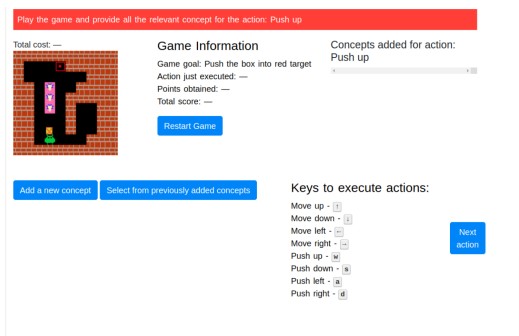

Figure 13: Screenshot from the survey done to collect sokoban concepts.

tive would be to reduce the overall burden placed on the user. While one could follow the basic interaction pattern laid out in this paper, namely, querying the user only when the current set of concepts proves to be insufficient, the use of such a reasoning framework would allow the system to pre-emptively collect concepts from the user that may be useful for future interactions.

**Confidence Threshold** Our system currently expects to be provided a confidence threshold that decides the minimum confidence that must be satisfied by any explanation provided to the user. Such a threshold may be provided by the stakeholders of the domain/system or the user. One could also determine when to provide explanations based on decision-theoretic principles. Since the confidence values are just probabilities, if the system has access to penalty values attached to getting an explanation wrong, then it can associate an expected value to an explanation (since after all the confidence is just the probability of the fact being true) and use it drive its decisions. This penalty value could be associated with both the user potentially making a mistake because of incorrect information but also could be related to a loss of trust from the system providing an incorrect explanation. One could also use our explanatory methods along with the explanatory confidence in the context of trust-aware decision-making systems like Zahedi et al. (2021).

**Ethical Implications** The use of confidence values makes sure that the system is not giving explanations to humans that it doesn't have high confidence in, thereby sidestepping many of the core issues related to post hoc explanations. One of the assumptions we have made in the main paper that allows us to do this is that the agent can correctly reason over its internal model. So if we can learn an equivalent representation for this internal model, then one could guarantee that the explanations are sound, in so far that the agent would only generate behaviors that align with the stated model components. However, this assumption may not always hold and the agent could generate behavior that may not conform to its internal models. Unfortunately, in such cases, we would need additional mechanisms to test whether the agent conforms to the identified model component. Otherwise, the explanation could lead to the human assigning undeserved trust to the agent. One scenario where the soundness of the agent's reasoning process doesn't matter are cases where the mechanism described in the paper is being used purely as a way to generate preference accounts Langley (2019), i.e., the system is purely trying to explain why one decision may be better than other in terms of its model of the task, regardless of how the agent generated the original decision. Such explanations could be helpful in cases where the agent and the user are collaborating to generate better solutions.

## A.12 STUDY INTERFACE SCREENSHOTS

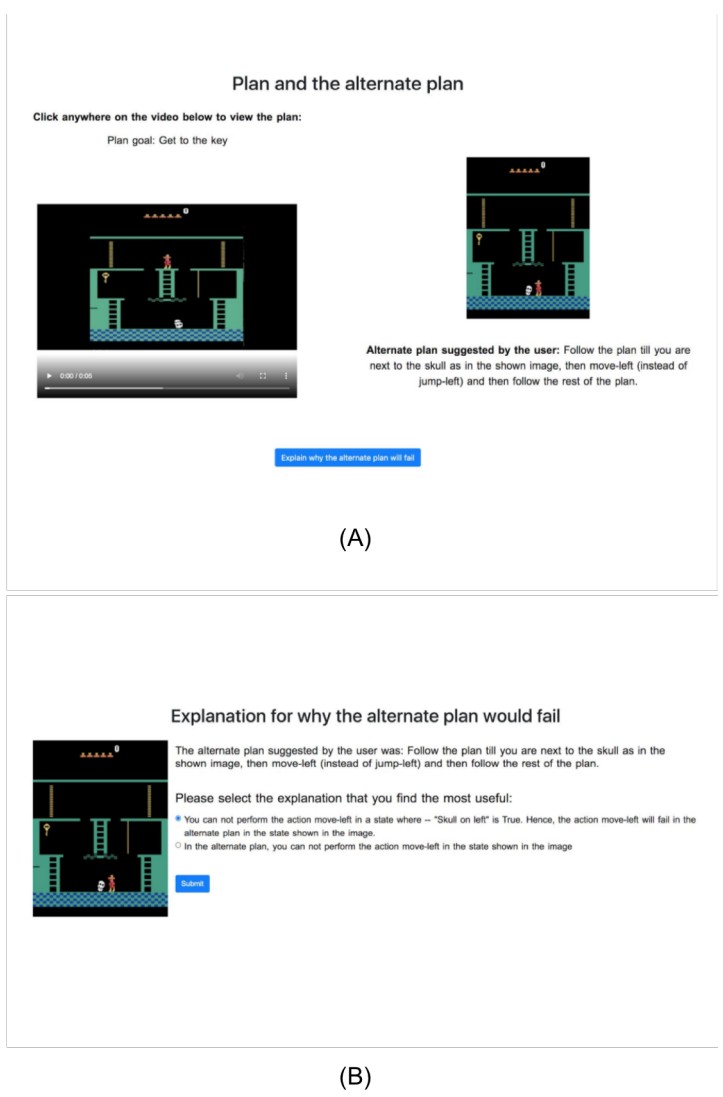

Figure 14: Screenshot from the study interface for H1 for precondition.

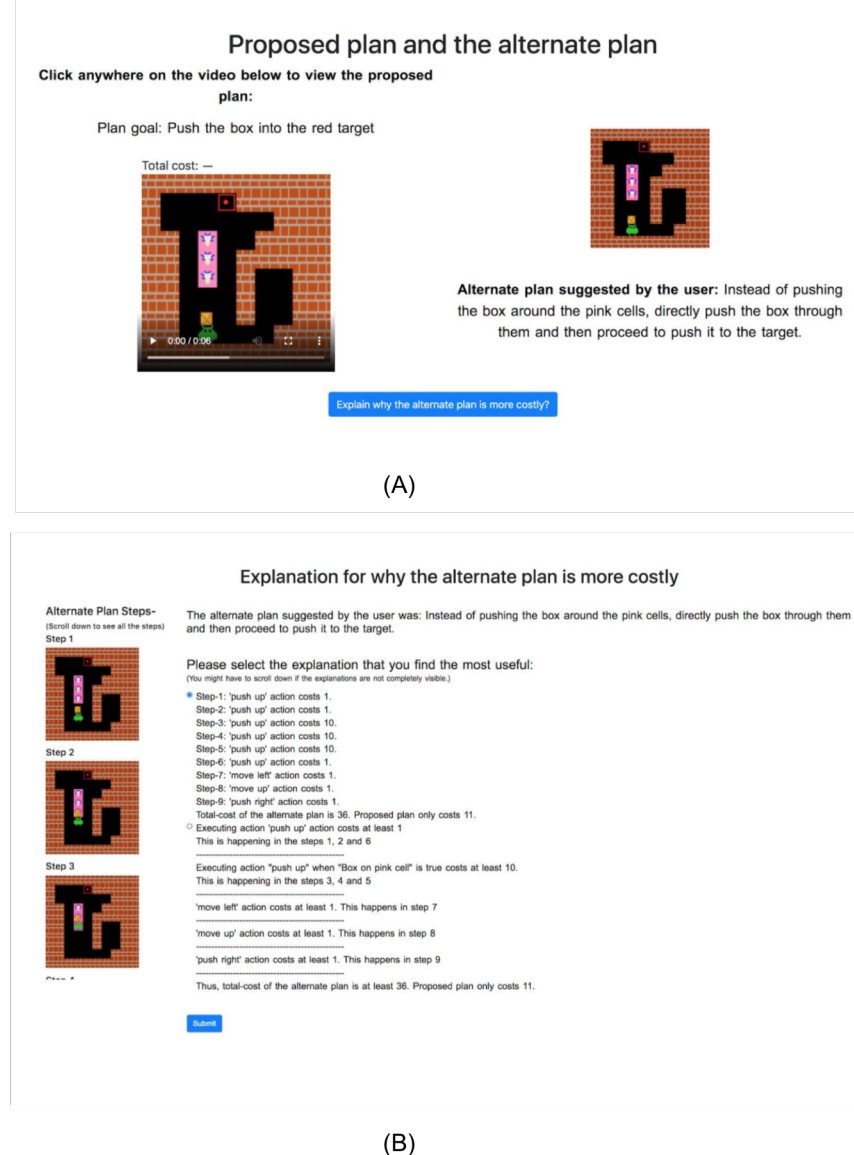

Figure 15: Screenshot from the study interface for H1 for cost function explanations.

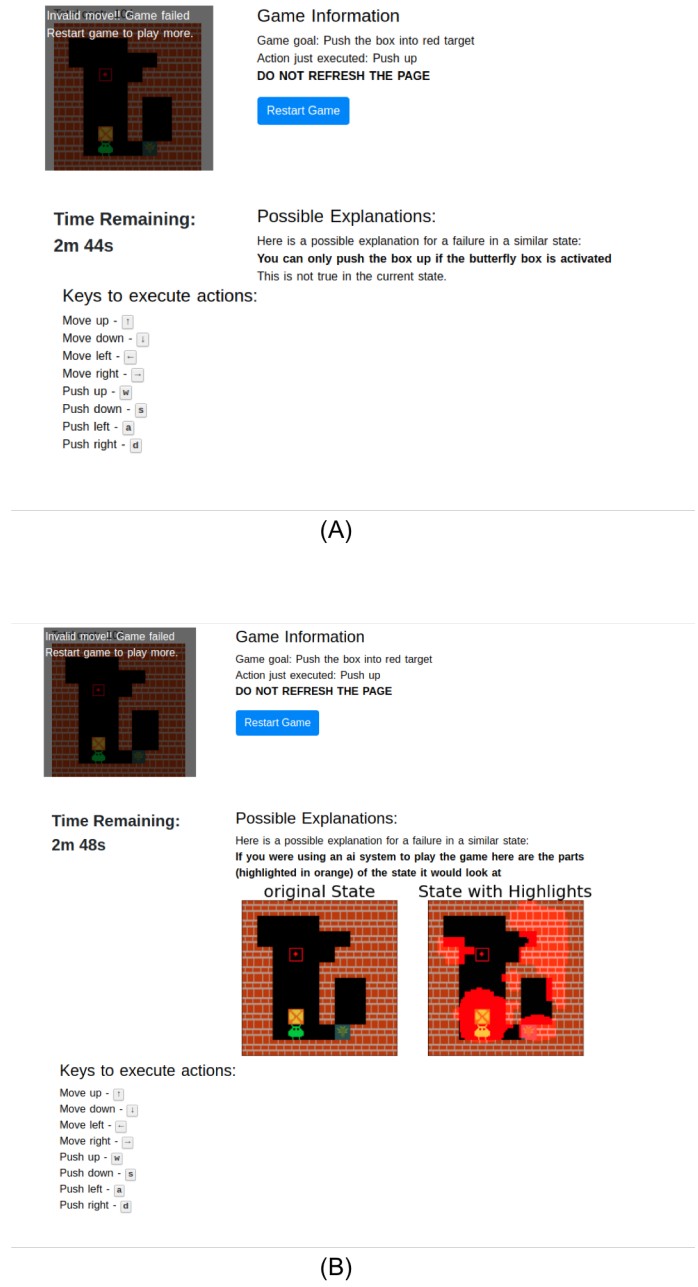

Figure 16: Screenshot from the study interface for H2.

