# OpenReview forum: "Bridging the Gap: Providing Post-Hoc Symbolic Explanations for Sequential Decision-Making Problems with Inscrutable Representations"
_ICLR.cc/2022/Conference — ICLR 2022 Poster_

### Official Review · Reviewer_ike6 · 2021-11-02

**Correctness:** 2
**Technical Novelty And Significance:** 2
**Empirical Novelty And Significance:** Not applicable
**Recommendation:** 5
**Confidence:** 4

**Main Review:**

Strength:
- AI interpretability is important;
- detailed appendix

Weakness:
significant presentation issues;
related work discussion;
model evaluation;
user study is not appropriate;

Detailed comments:

1.
I agree that AI interpretability is an important subject -- but unfortunately the poor presentation clarity in this paper makes it hard to assess the soundness of the claims.
Overall -- the text needs editing, there are no paragraph breaks, microscopic figure labels, and the discussion of related work is either scattered through the text or crammed in the end, instead of given in the introduction.

- Consider this 5-line-long sentence:
"As part of developing this system, we will also formalize the notion of local symbolic approximation for
sequential decision-making models, and introduce the notion of explanatory confidence (Section 5)
that captures the fidelity of explanations and help ensure the system only provides explanation whose
confidence is above a given threshold."

- Labels in Figure 5 require 300% zoom to see, and can not be read in print at all.

- The sentence "We had the smallest explanation likelihood for foil1 in screen 1 in Montezuma (with 0.511   0.001), since it used a very common concept, and their presence in the executable states was not strong evidence for them being a precondition."
Their presence sounds like several concepts, the sentence refers to one concept, and it is unclear which one?
The authors do a good job motivating the paper in the opening paragraph, and Figure 1, but unfortunately do not sustain this momentum through the remainder of the paper.

2. A comprehensive discussion of related work is missing, or scattered through the text, which makes it difficult to evaluate the novelty, contributions, and technical motivation of the current work.

3. The number of concept classifiers is reasonable, and the method of soliciting concepts through surveys is good, however computational model evaluation is not sufficient.
Evaluation of Montezuma's revenge used only 4 foils, evaluation of Sokoban uses only one foil.
One reasonable way to obtain such foils would be to obtain them from humans trying to play the game. The authors do use a study of users playing Sokoban to evaluate the effectiveness of explanations, so why not use the foils generated by users through play to evaluate the model as well.

4. The user study measures
 -- Experiment 1-- whether people prefer explanations that refer to the symbolic model components over more concise explanation
 -- Experiment 2-- whether the proposed explanations help people get a better understanding of the task, compared to a non-verbal highlighting of problem areas by a saliency map.

Study result reporting is unusual, it was very hard to understand what was actually done in the studies.
The authors report that 100 users participated in the study.
Further, Experiment 1 collected collected 5 responses per each of the 4 Montezuma foils, and 10 responses for each of the 2 sokoban foils, this means 50 responses, ...but the study is said to be within-subjects. Does this mean that each subject gave 50 responses, and responded multiple times to the same files?  However, Table 1 reports X out of 20 subjects in each condition - indicating that each subject responded to just one foil, and that there were 50 subjects in total.
After some reflection, I realized that 100 users participated in both Experiment 1 and Experiment 2.

Regarding Experiment 1, I am not convinced that this study appropriately evaluates the system.
The authors misunderstand the preference for concise explanations. The point of this preference is that people look for (1) simplest explanation to make sense of their observations (Occams razor), and (2) explanations that are pragmatic -- communicate more content with fewer words (e.g. Grieces maxims). Neither of those principles assumes that the communicator should sacrifice grammar, important causal information, or linguistic intelligibility, in order to say less.
The authors reduce this to an incorrect assumption that people should prefer to see less text.
For example, consider human players discussing two alternative plans. Player A proposes a plan that results in loss. Player B can decline by saying "death" or by saying "if we do this, we will loose because.." -- both explanations are technically correct, but one of them omits casual information. Player A could also say "death fall" - which presumably includes causal information, but is given in a less intelligible form.
The value of concise explanations is to reduce the listener's cost (processing the answer) while maximizing reward (relevant information). Each experiment needs to clearly state what the subjects did.

Regarding Experiment 2, the method of having human players play the task and measuring the number of steps they take is a good one, but the presented analysis has several issues.

 - "we collected 60 responses but had to discard 12 due to them missing to self-report that they looked at the explanation."
This implies that 12 subjects did not find the explanation useful, not that they should be discarded. Human players may prefer to "figure out" a game by active learning -- posing and testing their own hypotheses about how the game works, instead of reading about it -- this is a common human behavior, and should not be considered invalid.

- Is the saliency map explanation system known to actually known to be effective in the given Sokoban task variant? The illustration in Figure 10 implies that it is not really helpful, with a large area of the map being highlighted at the same time.
It would be good to see how many steps do people take in a "no explanation" condition. Further, this information needs to be presented with 95% confidence intervals, and would be more readable if presented as a bar plot rather than a table.

- Were the Sokoban study subjects given task instructions?




**Summary Of The Paper:**

This paper proposes a symbolic model implemented alongside a planning AI, which can respond to human  queries about why one plan is taken by the AI, and not another.  The system takes as input a user concept vocabulary, where each concept is implemented by binary classifier indicating whether the proposition is present. This concept vocabulary needs to be initially solicited from the user -- either through user surveys or designed by researchers.
The output of the classifiers is noisy, which is interpreted as confidence over the proposed explanation. The system then uses a symbolic PPDL model with the concept vocabulary to answer queries. The system can issue two types of answers about why a proposed plan is inadmissible: (1) a pre-condition for a given action is missing or (2) the proposed plan is more costly than the chosen plan.

To identify if a concept is a precondition for action A, the system samples a number of states from the model where A is executable. If concept is always present, then it is a pre-condition. Samples are generated by a random walk from one of the states in the model's given plan. In the evaluated problems games the sampling algorithm needs ~500 samples to accurately identify the missing pre-condition.
To identify the cost associated with executing a proposed action, a heuristic is used, such that tries to sample the minimal cost when executing the action given the preconditions. (This is not clearly explained )

A small user study was used to evaluate whether the subjects find the proposed explanations helpful.


**Summary Of The Review:**

I am not convinced that this paper is ready for publication in ICRL. There are many good things to this paper -- such as a beautiful motivating example on the first page, and a detailed technical appendix -- but it still needs more work to be of interest to the broader ICRL community. The paper has significant presentation issues, which make it hard to evaluate the soundness of the claims, theoretical significance and novelty.
The described model evaluation, and user study, are insufficient to support the claims.

---

> ### Author Response · Authors · 2021-11-11
> **Response to Reviewer Comments**
>
> We thank the reviewer for their suggestions, in particular regarding the structure of the paper. The current structure of the paper is mostly informed by previous comments we had received from earlier conferences. We will make sure to incorporate the reviewer’s suggestions into the paper and hope to submit a revised version in a few days. Below we have provided answers to some of the specific questions the reviewer had
>
> Computational Evaluation: The reason for the smaller number of foils in Sokoban variants was due to the fact that we needed to create a simple enough domain that we could have a wide variety of users interact with and provide useful concepts for. This meant reducing the number of possible failure points to a few specific ones. While each of the players may have provided a slightly different plan, their failure explanations (in regards to validity or suboptimality) would have stayed the same. Hence we chose to focus on a single exemplary foil per variant.
>
> Hypothesis1 User Study: This hypothesis is meant to be a baseline experiment that presents a first-order validation of the information we generate. While we agree with the reviewer that intuitively, since we are generating causal information, people would find it useful. We still chose to test it explicitly as we don’t know of any work that demonstrates it in explainable reinforcement learning settings. Part of the issue comes from the fact that there doesn’t exist comparable automated methods that can generate symbolic causal information. This also limits the choice of baselines we can use for evaluating our system. We didn’t include a handwritten explanation as a baseline (for example "death fall") since this would bias the study setting and possibly introduce other confounders.
>
> Filtered out Candidates: The 12 participants who were filtered out, were done so based on their response to the question “Did you see any explanations?” which was a yes or no question. This was just there to see if participants were paying attention to the study. Ideally if they read the question correctly, people should only say no to this question if they didn’t pay attention to the game page at all, just let it expire or never tried a failing action. It is possible that someone could have said no to the question, because the information didn’t meet their expectations about a good explanation, but treating all 12 participants as a coherent group would not be helpful.
>
> Saliency Map Choice: As far as we know, Sokoban is not a widely used domain in explainable reinforcement learning. Which makes sense given its complexity and the need to do long term planning. We chose the specific method as it was the most robust one we were aware of and was also demonstrated to be effective by [1] (also mentioned in the paper). Also the confidence intervals are provided in Table 2 (b) (the interval provided was not standard deviation but confidence intervals and we will make sure to mention this in the paper).
>
> Sokoban Task Instructions: The study participants were provided with information about the objective of the game and the controls (the actual instructions can be found in the supplementary zip file in /ICLR_22_3903/Study_Website_Pdfs/h3_sokoban_website.pdf). They were expected to learn the exact rules of the game by playing, which allows for the possibility of more mistakes and hence more chances for the system to provide explanations.
>
> [1] Zhang, Ruohan, et al. "Machine versus Human Attention in Deep Reinforcement Learning Tasks." arXiv preprint arXiv:2010.15942 (2020).

---

> > ### Comment · Reviewer_ike6 · 2021-12-04
> > **Thank you for the responses**
> >
> > Thank you for the responses. In the revised manuscript the readability has indeed improved. I'm happy to rise my score by a point.

---

### Official Review · Reviewer_mLyQ · 2021-11-02

**Correctness:** 3
**Technical Novelty And Significance:** 3
**Empirical Novelty And Significance:** 3
**Recommendation:** 8
**Confidence:** 4

**Main Review:**

Explainability is a highly relevant field within machine learning as more and more models find applications in critical areas. While there’s been substantial amount of investigation into explainability for classification models, generalization to other frameworks remains lacking. This paper does a good job of formalizing what kind of explanations might make sense for sequential decision-making problems, and it is easy to read and follow. The development is systematic, and the kind of explanations proposed appear sensible to me as an end-user. There is also extensive background and supplementary material provided in the appendix to support and reinforce the ideas. I think this is a promising albeit quite preliminary work in an ill-explored area.

My main concern is the limited nature of the evaluation section. It appears that the assertions are largely substantiated via a handful of examples and it’s unclear to me how these ideas might generalize. Needing to handcraft classifiers, learning preconditions for feasibility of actions via sampling, assuming certain variables to be independent (e.g. non-precondition concepts), imposing a loose generative model on the process, all appear somewhat fragile. While the appendix touches upon some of these weaknesses, it’s not entirely convincing without more exhaustive evaluation. Relatedly, the results of the human study with the subjects preferring concept-level explanations over saliency-based ones are appreciated but unsurprising. I wonder how the proposed method will compare against another parsimonious account of relevant concepts (e.g. skulls kill, falling off the edge kills, can’t go through crabs).

**Summary Of The Paper:**

In this paper, the authors present a method for justifying an agent’s policy in a sequential decision-making task when the human proposes specific counterfactuals. The authors first represent the state of the system with a set of intuitive classifiers, and then explain why a counterfactual (termed foil), would fail or cost more than the agent’s policy based on this classifier representation. For example, identifying that “move left” leads to a failure state in a game when the current state has the classifier “platform_edge_on_left” = True, as one its descriptors. Similar ideas extend to identifying an overly costly foil policy. The authors develop how such state characterization can be constructed, how feasibility of actions can be learned from sampling, and how a simple Bayesian formalism can be used to quantify uncertainty arising from sampling and other assumptions. They go on to show how the proposed idea works for two game settings, as well as how these explanations are preferred by human subjects over saliency based ones.

**Summary Of The Review:**

Overall, I think this is a good paper with the potential to spark more work in the area. However, the evaluation section can be made stronger to really convince the reader of the merit of the method. In its current draft, I am somewhat inclined for the paper to be accepted.

---

> ### Author Response · Authors · 2021-11-11
> **Response to Reviewer Comments**
>
> We thank the reviewer for their positive comments and suggestions.
>
> Evaluation: None of the classifiers were handcrafted, rather we used specific existing methods to learn the classifier. In fact for the Sokoban all the concepts used in the experiments were ones that external participants provided to us after interacting with the task.
>
> Form of Explanatory Messages: We used the current form of explanation, as they can be generated easily through template filling. We avoided using more direct handwritten explanations as they could bias the results. Introducing a natural language generator which generates more parsimonious messages would definitely be a natural next step for this work.

---

### Official Review · Reviewer_wvCE · 2021-11-04

**Correctness:** 4
**Technical Novelty And Significance:** 3
**Empirical Novelty And Significance:** 3
**Recommendation:** 6
**Confidence:** 2

**Main Review:**

Explanation is contrastive [1] -- because in order to answer the question to why agent does X, there can be many different reasons, and only by contrast, we can truly perform causal selection (selecting the causes that explain the difference).

Strengths:
1. The paper is very well-written, but Section 3 and the algorithm can be a bit unclear to people unfamiliar with symbolic logic.
2. I really like Section 5, where the confidence of explanation is quantified through a concept classifier, and it can additionally be incorporated into the search algorithm.

Weakness:
1. The need for a foil agent seems like a very big impediment of allowing the methodology proposed in this paper to have real-life application/impact. Collecting concepts from user seems very doable, but soliciting foil actions/agent from users is non-trivial. This is not a strong criticism on the paper, because the authors can suggest that the real explanation people need from an agent is: "Why do you do X instead of Y?"  -- this format of question is exactly what this paper captures.
2. I really don't think game is the best domain -- because rarely would people want a game agent to justify its action when playing a game. RL policy is used in many other domains such as recommendation systems, healthcare, education -- in there, action space is more limited, and the need for policy to justify itself does exist. I wonder if the authors can comment on that and work through a concrete example, say in healthcare (ICU) + RL setting, what kind of concepts might be used in there, what could be the performance of the concept classifier, what does a foil action/agent look like, and whether the paper's method can provide satisfying explanations in a setting like [2] (no need to run actual experiment, a response with some worked out examples would be great!)
3. In Section 2, the authors said "We will focus on models where the preconditions are represented as a conjunction of propositions." -- this is never justified or explained. I can only assume there is some justification on why conjunction of propositions can perfectly represent pre-conditions. And why are pre-conditions ideal explanation candidates? This might seem very obvious to the authors -- but stating the justification can be important for future work to either expand on authors' idea.

Overall, I think this paper contributes nice ideas to the field of XAI in RL. I wish the domain is not game -- but clearly, game is the easiest domain to focus on, and concepts can be easily acquired. I have some concerns on the real-world applicability of the methodology. However, the paper is well-written, and the information presented is a nice contribution to the discussion of XAI. I'm happy to raise my score if the authors address my concern.

[1] Hesslow, Germund. "The problem of causal selection." Contemporary science and natural explanation: Commonsense conceptions of causality (1988): 11-32.
[2] Komorowski, Matthieu, et al. "The artificial intelligence clinician learns optimal treatment strategies for sepsis in intensive care." Nature medicine 24.11 (2018): 1716-1720.

**Summary Of The Paper:**

The paper presents a search procedure with concept classifiers to generate symbolic explanations (in terms of preconditions and costs) to justify an agent's action w.r.t. a foil agent. The authors conducted user study to show the usefulness of their method.

**Summary Of The Review:**

The paper is well-written and the methodology is novel. However, XAI is most needed in critical domains like recommendation systems, healthcare, and education. Would be good to see some discussion in these areas.

---

> ### Author Response · Authors · 2021-11-11
> **Response to Reviewer Comments**
>
> We thank the reviewer for their constructive feedback and are happy that the reviewer found the methods novel. Below we have replied to some of the concerns raised by the reviewer
>
> Games as Evaluation Domains: The reason we chose a game domain as the test bed was not because of availability of concepts, but rather because it's easier to run larger user studies.  Using a game domain generally lets us run studies with a wider or more diverse population (as we don’t have to rely on domain experts) or need to rely on a long training phase where we teach the participants about the task. We would imagine getting concepts in more specialized domains would be a bit easier, due to larger availability of documentation and we could potentially acquire them from domain experts. In the sepsis example, the concepts would include the various medical markers the doctor considers valid for the diagnosis and the action space would correspond to various dosage actions.
>
> Use of Conjunctive Precondition: This assumption allows us to simplify our explanation generation process without compromising on the generality of the method (Proposition 3 in the appendix establishes the generality). By assuming a conjunctive precondition, we require that every literal part of the formula has to hold in a given state for the action to be executed. This means we only require finding a single failing concept to explain the failure of the entire action. We will make sure to include a discussion of this choice in the main paper.

---

### Official Review · Reviewer_C3CZ · 2021-11-04

**Correctness:** 4
**Technical Novelty And Significance:** 3
**Empirical Novelty And Significance:** 4
**Recommendation:** 10
**Confidence:** 4

**Details Of Ethics Concerns:**

The paper includes user studies. They report the protocal and the involvement of local body of ethics.

**Main Review:**

# Thank you

Thank you for your comments and changes. I have a suggestion:
Please add a short paragraph discussing what would happen if the users only mention part of the crucial concepts.
This is related to a transversal challenge on the method applicability being sensitive to the concepts that the users provide.

I also find it interesting to consider the other extreme: what if the user were providing the perfect set of concepts.
What would be happening in that cases? It seems the algorithm would be dealing with some form of model-based seq decision making.

# Older main review
The paper emphasizes that the method is not trying to learn a full symbolic representation for the problem. Besides, given the intrinsic difficulty of explanations of sequences, the method should aim to high precision. I keep in mind that there are no solid alternatives for this setting.

A potential weakness is the requirement of the concepts. I wonder what the alternatives are. All explainability setting convey their meaning in some way. In this case, the concepts are on the side of the users of the method. In this case, user experiences designers have tools that they can exploit in multiple ways. It should be possible to use abstract concepts that correspond to user interface elements, as far as they can be visualized later. The proposed method is orthogonal to that. Providing concepts is a relatively cheap label for sequential decision-making. Compare that with the complexity of labelling videos. Besides, vocabulary seems a minimal form of communication.

The general idea is both interesting and simple for those familiar with STRIPS planning. The appendix considers more detailed technical scenarios. However, a main important issue is explained in the body of the paper: the observations of concepts it bound to be noisy. The graphical model –explained in detail in the appendix– allows to provide probability estimates to guide the selection of explanations. The estimation using the graphics model is similar to some inferences on hidden Markov models. That's not casual since we are talking about sequences of actions and hidden variables.

I am pretty satisfied with this revision of the paper, so I don't have many questions or comments. See some below.


page 2: *We will denote the state formed by executing action a in a state s as a(s). We will focus on models where the preconditions are represented as a conjunction of propositions.*
What about the action effects? What do they do?

page 4: *Figure 2: Algorithms for identifying (a) missing precondition and (b) cost function*
I understand what the algorithms do. However, the figure could be more clear. For instance, what's poss_prec_set?

page 5: *The worst-case time complexity of this search is linear on the sampling budget*.
Would you please comment more clearly about the cost/resources of producing explanations? Both in preparation and for a particular instance.



**Summary Of The Paper:**

(NB: I reviewed a previous version of this paper. I was in favour of its acceptance. In my opinion, this version is far more clear and addresses the most salient concerns of the reviewers, including me. I'll keep this review self-contained and refer to this submission).

The paper proposes a methodology for providing post-hoc explanations in sequential decision settings without assuming a model of the environment. Usual methods for explainability do not translate well into sequential decision settings, as the explanation is related to the executed actions. In particular, what's the natural form for such explanations? A simple but useful query is to ask about modifications to the plan retuned by the AI agent. These are called constrastive explanations. Those modifications could be a) better, b) lead use some illegal action execution, c) be more expensive.

The method creates a symbolic approximation around the current situation. The approximation allows to show the user a single concept that is missing and makes an action fail, or to increase the cost. The validation with users reports that they are finding them usefull. A more indirect test, less sensitive to the bias of self-report, is whether they can solve a task faster. Those results are also positive.

**Summary Of The Review:**

In summary, I recommend the acceptance of the paper. The idea is very simple and can be combined with efforts both on explainability and neurosymbolic methods that offer trust to the users.

The paper presents a strong combination of minimal information used to ground the explanations with learning and reasoning methods to provide adaptation and coherence. The use of planning allows to produce small explanations in the most critical parts.

If anything, the weakest point is that the notion of locality might be hard to control. In some cases, the reason for an action not being possible might be further away. Another round of interaction with the users might allow to discover for what concepts it might be useful to look further. For instance, a video game might feature a switch that should be open to achieve some effect.

---

> ### Author Response · Authors · 2021-11-11
> **Response to Reviewer Comments**
>
> We thank the reviewer for their comments and we are happy that the reviewer found the changes to the paper helpful. Here are responses to specific questions/comments the reviewer had, and we will also update the paper to reflect that
>
> Form of action effects: Effects of an action describe what happens when an action is executed. In deterministic settings the effect of an action in a given state is always conjunctive, we will make sure to mention this in the paper.
>
> Missing information about Algorithms: Thank you for catching that, poss_prec_set is the hypothesis set we maintain throughout the search in regards to the possible preconditions. We will make sure to include more details about the pseudo code notations in the text.
>
> Resource Usage: The worst case complexity assumes the samples are collected sequentially and in many practical cases this could be improved by parallelizing the sampling process (by running parallel random walks). The only other possible complexity is that associated with sampling an outcome from a simulator, of course this is very much dependent on the specific simulator implementation. For example, for Montezuma’s revenge collecting 100 samples from level 1 took around 60 secs. We will make sure to include the average time taken for all the domains considered in the paper.

---

> ### Author Response · Authors · 2021-12-02
> **Response to post-rebuttal comments**
>
> Thank you for the suggestion, we will add a discussion section on how we see the longitudinal interaction process evolving. As mentioned in the paper, our method allows for both cases, in that if the human gave all the required concepts it will generate the most likely explanation and also detect scenarios where the current concept set is insufficient. The latter scenarios will be characterized by either empty hypothesis set after the sampling or by a low confidence explanation (if all the concept classifiers are extremely noisy). In such cases, the method could potentially query the human for more concepts. The discussion section in the appendix does contain a small sketch of how such an interaction could occur (which we will combine with this discussion). The reviewer is right that rather than viewing it as an ad-hoc procedure, one could model it using a meta decision-making process. One where we attach a cost to querying humans for more concepts and a high penalty to providing the human a wrong explanation. This is definitely an exciting future direction for the work. It would also be interesting to figure out if there are ways we can model the possible queries human could ask (say in terms of possible future states or problem instances they may see), so the decision-making framework can reason about whether it should pre-emptively collect concepts about certain parts of the state space, even before the human asked questions relevant to it.

---

### Author Response · Authors · 2021-11-18
**Changes in the Revised Version**

Here are the list of changes we made in the revised version, most of the changes are also highlighted in blue in the paper

1. We have moved the related works section ahead and also consolidated the discussion about related works

2. We have updated the sentence in the evaluation section to remove any confusion that the selectivity property is completely characterized by linguistic terseness. We never meant to imply that, after all, our results show otherwise.

3. We have added a note about the use of variable poss_prec_set

4. We had added a note in the appendix (A8) about average sampling time for the domains

5. We have tried to improve the readability of the paper at multiple points, including improving the size of the font in figure 5, breaking up long sentences and adding paragraph breaks.

---

### Decision · Program_Chairs · 2022-01-20

**Decision:**

Accept (Poster)

**Comment:**

The paper gives a method for generating contrastive explanations, in terms of user-specified concepts, for an agent in a sequential decision making setting.

The reviewers found the paper to be a strong contribution to explainable AI and RL. There were some concerns about the writing, but the revisions have addressed most of these.

Overall, I am delighted to recommend acceptance. I urge the authors to incorporate the feedback in the reviews in the final version.